# Base Models for Parabolic Partial Differential Equations

**Xingzi Xu**[*1]      **Ali Hasan**[*†1,2]      **Jie Ding**[3]      **Vahid Tarokh**[1]

[1]Department of Electrical and Computer Engineering, Duke University, Durham, North Carolina, USA
[2]Machine Learning Research, Morgan Stanley
[3]School of Statistics, University of Minnesota, Minneapolis, Minnesota, USA

## Abstract

Parabolic partial differential equations (PDEs) appear in many disciplines to model the evolution of various mathematical objects, such as probability flows, value functions in control theory, and derivative prices in finance. It is often necessary to compute the solutions or a function of the solutions to a parametric PDE in multiple scenarios corresponding to different parameters of this PDE. This process often requires resolving the PDEs from scratch, which is time-consuming. To better employ existing simulations for the PDEs, we propose a framework for finding solutions to parabolic PDEs across different scenarios by meta-learning an underlying base distribution. We build upon this base distribution to propose a method for computing solutions to parametric PDEs under different parameter settings. Finally, we illustrate the application of the proposed methods through extensive experiments in generative modeling, stochastic control, and finance. The empirical results suggest that the proposed approach improves generalization to solving new PDEs.

## 1 INTRODUCTION

In this work, we propose and study a particular type of neural structure that can adapt rapidly to tasks associated with parabolic partial differential equations (PDEs). Parabolic PDEs are a standard mathematical framework for describing the evolution of different processes. Applications are evident in various fields, such as probabilistic modeling and mathematical finance [Pardoux and Râșcanu, 2014]. For instance, the probability density functions (PDFs) in generative modeling with diffusion processes satisfy a parabolic

PDE known as the Fokker-Planck equation. In continuous stochastic control problems, the optimal policy satisfies the Hamilton-Jacobi-Bellman equation. In finance, the Black-Scholes equation describes the price of a derivative. Due to the numerous applications, there is a general need to describe such processes under different *boundary conditions* and *parameters*. In the probabilistic modeling example, we may consider a scenario where we wish to sample from multiple related distributions while using their shared features to accelerate training. In this case, different diffusion processes can correspond to distinct parameters of the Fokker-Planck equation.

Parabolic PDEs have a particular structure that allows for efficient computation of their solution in high dimensions using Monte Carlo techniques [Pardoux and Râșcanu, 2014]. The Feynman-Kac formula formalizes this for linear parabolic PDEs through a connection between an expectation over sample paths and the solution to the corresponding PDE [Särkkä and Solin, 2019, Chapter 7.7] with the nonlinear extension following a similar argument [Fahim et al., 2011]. However, the formula requires sampling solution paths of a stochastic differential equation (SDE) with parameters corresponding to the PDE. This process can be time-consuming due to the large number of sequential samples needed, and it requires sample paths to be *resampled* for different parameters of the PDE. We instead build upon this idea to approximate the PDE solutions by recycling Monte Carlo samples used in computing the solutions while mitigating the instabilities inherent in the direct application of importance sampling. This idea leads to a base model (or meta model) that can swiftly adapt and extend to different tasks (e.g., solving different PDEs or sampling distinct target distributions). We borrow techniques from two research disciplines: the first related to meta-learning and the second related to operator learning [Finn et al., 2017, Lu et al., 2019, Jin et al., 2022]. Rooted from meta-learning, we want to modify the base model to unseen tasks with lower training effort. Related to operator learning, we want to define an operator using the base model that maps pa-

---

[*]Equal contribution.
[†]Corresponding author.

*Accepted for the 40th Conference on Uncertainty in Artificial Intelligence* (UAI 2024).

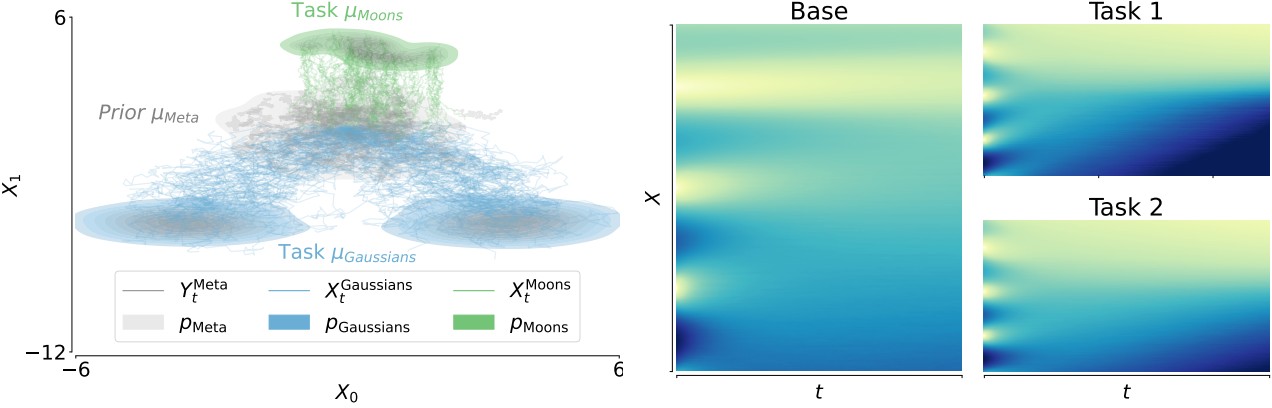

(a) Maximizing likelihoods of target distributions $p_{\text{Gaussians}}, p_{\text{Moons}}$ using a shared base distribution $p_{\text{Meta}}$.

(b) Solving PDEs with different parameters (Task 1 and 2) by reusing the base solution.

Figure 1: Schematic of the proposed procedures for maximizing functions (1a) and solving PDEs with different parameters (1b). In both scenarios, we reuse a meta-learned base parameterization across different tasks. (1a) illustrates sampling target densities using a task-specific diffusion for each density. (1b) illustrates two solutions to linear parabolic PDEs simulated with the same stochastic process based on the proposed method.

rameters and boundary conditions to solutions. The outline of the paper is as follows: We first describe the importance sampling framework in section 2.3; we then discuss the issues of directly applying importance sampling and introduce the proposed meta-learning approach in section 3.1. In section G, we describe a lower bound for applying the framework to maximization tasks, such as maximizing the likelihood in generative modeling. In Section 3.3, we generalize the computation for maximization tasks and present a neural operator that allows computing solutions to PDEs at a reduced cost. Figure 1 illustrates an overview of these concepts where: on the left, we apply the proposed framework to sample from different target distributions corresponding to $p_{\text{Moons}}, p_{\text{Gaussians}}$ using a shared base model $p_{\text{Meta}}$; and, on the right, we apply the operator framework to solve two parabolic PDEs (Task 1 and 2) with different parameters by reusing the original simulated stochastic process (Base). [1]

**Contributions** To summarize, our main contribution is a framework for establishing a base model for reuse across problems associated with parabolic PDEs. To that end,

- We describe the base model and a corresponding meta-learning framework for solving maximization problems associated with parabolic PDEs;
- We use the meta-learning framework and propose a neural operator to approximate the solution of parabolic PDEs under different parameter settings;
- We analyze the convergence of the operator over parametric classes of functions;

- We evaluate the methods in different experimental settings, including synthetic and real-world examples.

## 1.1 RELATED WORK

We will discuss the related work on meta-learning techniques for PDEs and operator-learning methods for PDEs. Psaros et al. [2022], Chen et al. [2022], Liu et al. [2022] consider applying the MAML framework to physics-informed neural networks (PINNs) [Raissi et al., 2019b]. In these approaches, the authors learn a meta-parameterization of PDEs for efficient optimization to estimate PDEs under new parameters. Huang et al. [2022] considered a latent variable model fine-tuned to obtain the solution at different regions of the parameter space and domain based on the PINNs framework. Chen et al. [2022] uses hypernetworks for multi-task learning with parameterized PDEs, which also focuses on low-dimensional settings.

On the second front are neural operator architectures used to solve PDEs. Operator-learning methods map sets of parameters to solutions of PDEs by estimating the operator that describes this transformation. The work of Lu et al. [2019] proposed DeepONet, which uses the result that a neural network with a single hidden layer can approximate any nonlinear continuous operator, and follow-up work of Li et al. [2020] considers learning a mapping in Fourier space with improved empirical performance. Gupta et al. [2021], Wang et al. [2021b], Li et al. [2022], Zheng et al. [2022] propose additional variants on this operator framework. However, these methods require first generating a dataset of parameter and solution pairs by solving the PDE according to standard techniques, which can be expensive. Wang et al. [2021b] attempts to circumvent this issue by including the PINN

---

[1]Github repository to this paper is: `https://github.com/XingziXu/base_parabolic.git`

loss in the operator learning framework. Additionally, Hu et al. [2023] considers learning PINNs for high-dimensional PDEs by decomposing a gradient of PDEs into pieces corresponding to different dimensions. However, various issues associated with optimization persist with the PINNs framework, e.g., in Krishnapriyan et al. [2021], Wang et al. [2021a], making its application difficult with certain PDEs. Finally, several methods consider the stochastic representation used in this paper. Berner et al. [2020], Richter and Berner [2022], Glau and Wunderlich [2022], Zhang et al. [2023] consider various forms of regression-based techniques wherein the stochastic representation is repeatedly sampled for different parameter values and regressed to a sufficiently expressive function approximator (usually a neural network). However, this incurs significant training costs due to the required sampling at each iteration in training and a feature that the proposed methods in this work explicitly circumvents. Han et al. [2018] consider solving semi-linear parabolic PDEs by regressing the stochastic representation of the solution to the neural network. A hybrid approach that considers both PINN losses and the stochastic representation of PDEs was considered in Nüsken and Richter [2021] for the single parameter case. These methods, however, only provide solutions for PDEs for one particular set of parameters rather than a family of parameters.

To address these gaps in the literature, including the lack of scalability to high dimensions, requiring many solution pairs for training, and requiring computing new solutions for different parameters, our framework exploits the stochastic representation of parabolic PDE with an importance sampling technique to transform the learned solution to one with distinct parameters.

# 2 STOCHASTIC REPRESENTATIONS OF PARABOLIC PDES

We will now describe the key ingredients of the proposed method and how to apply them in the proposed framework. Throughout the paper, we will refer to the base model interchangeably as a meta model.

## 2.1 PARABOLIC PDES

Consider a domain $\mathcal{D} \subset [0, T] \times \mathbb{R}^d$ such that the solution $p(t, x) : \mathcal{D} \to \mathbb{R}$ of the PDE is defined. A linear parabolic PDE is of the form

$$\frac{\partial p}{\partial t}(t, x) = \nabla p(t, x) \cdot \mu(t, x)$$
$$+ \frac{1}{2}\mathrm{Tr}(\sigma\sigma^\top(t, x)(\mathrm{Hess}_x p)(t, x)) - r(t, x)p(t, x), \quad (1)$$

with boundary condition $p(0, x) = p_0(x)$. The function $\mu(t, x) : \mathcal{D} \to \mathbb{R}^d$ is referred to as the *drift function* and the function $\sigma(t, x) : \mathcal{D} \to \mathbb{R}_+^{d \times d}$ where $\sigma\sigma^\top$ is semi positive

definite for all elements in $\mathcal{D}$ is referred to as the *volatility function*. The function $r(t, x) : \mathcal{D} \to \mathbb{R}$ is sometimes referred to as the *growth term*. For all functions, we require the usual conditions on $\mu, \sigma, r$ such that (1) is uniformly parabolic (e.g. see Evans [2022, Chapter 7.1]). $\mathrm{Hess}_x p(x, t)$ denotes the Hessian of $p$ with respect to $x$. For the remainder of the text, we will focus on the interplay between the functions $\mu, \sigma$, and $p_0$, as we are interested in how they can be easily updated to solve different tasks.

## 2.2 FEYNMAN-KAC METHOD

Solving (1) in high dimensions generally requires using a Monte Carlo method to alleviate the curse of dimensionality. In particular, the Feynman-Kac method provides such a mechanism:

**Lemma 1** (Feynman-Kac method Särkkä and Solin [2019]). *Let $X_t$ satisfy the following Itô diffusion:* $\mathrm{d}X_t = \mu(t, X_t)\mathrm{d}t + \sigma(t, X_t)\mathrm{d}W_t$. *Then, the solution of the PDE in* (1) *is*

$$p(t, x) = \mathbb{E}\left[p_0(X_t)\exp\left(-\int_0^t r(s, X_s)\mathrm{d}s\right)\bigg| X_0 = x\right].$$
$$(2)$$

While the Feynman-Kac method applies to various types of parabolic equations, we present it in the case of solving Equation (1) with an initial condition to simplify the exposition.

## 2.3 CHANGING PARAMETERS

From (2), changing the parameters of the PDE requires sampling new sample paths of $X_t$. We aim to decrease this sampling burden by reusing the existing samples for different parameters. To do this, we consider an application of Girsanov's theorem:

**Definition 1** (Likelihood Ratio [Särkkä and Solin, 2019]). *Let $X_t, Y_t$ be two Itô diffusions satisfying*

$$\mathrm{d}X_t = \mu^{(1)}(t, X_t)\mathrm{d}t + \sigma(t)\mathrm{d}W_t,$$
$$\mathrm{d}Y_t = \mu^{(2)}(t, Y_t)\mathrm{d}t + \sigma(t)\mathrm{d}W_t$$

*with laws $\mathbb{P}_{X_t}, \mathbb{P}_{Y_t}$. Define $\Sigma = \sigma\sigma^\top$ and $\delta\mu(s, x) = \mu^{(1)}(s, x) - \mu^{(2)}(s, x)$ and assume $\mathbb{E}\left[\frac{1}{2}\int_0^t \delta\mu(s, x)\mathrm{d}s\right] < \infty$. Then, the* exponential martingale *gives the likelihood ratio of the two processes:*

$$\frac{\mathrm{d}\mathbb{P}_{X_t^{(1)}}}{\mathrm{d}\mathbb{P}_{Y_t^{(2)}}} := \exp\bigg(-\frac{1}{2}\int_0^t \delta\mu(s, x)^\top\Sigma^{-1}\delta\mu(s, x)\mathrm{d}s$$
$$+ \int_0^t \delta\mu(s, x)^\top\Sigma^{-1}\mathrm{d}W_t\bigg). \quad (3)$$

We will drop the superscript in $\frac{d\mathbb{P}_{X_t^{(1)}}}{d\mathbb{P}_{Y_t^{(2)}}}$ except for cases where specifying the drift index is necessary. Definition 1 relates to Girsanov's theorem [Särkkä and Solin, 2019] and note that the $X_t, Y_t$ originates from the same filtration $\mathcal{F}_t$ generated by the Brownian motion $W_t$. The likelihood ratio facilitates computing the expectation of a function of $X_t$ by using generated samples of $Y_t$. Specifically, it holds that

$$\mathbb{E}\left[p_0(X_t) \mid \mathcal{F}_t\right] = \mathbb{E}\left[p_0(Y_t)\frac{d\mathbb{P}_{X_t}}{d\mathbb{P}_{Y_t}} \mid \mathcal{F}_t\right] \qquad (4)$$

meaning that to change the PDE parameters $\mu, \sigma$, we only need to compute a transformation of the existing sampled paths rather than resampling from scratch. A standout example is the case of Brownian motion when $\mu^{(1)}(t, X_t) = 0$, where we can sample $N$ Gaussian random variables with variances depending on $\sigma$ and $t$ (i.e., independent of state) and reuse the sample path for different $\mu^{(i)}$. Through Itô's lemma, we can apply a function $f$ to the sampled Brownian motion to find a new sample path that has non-unit volatility[2] and depends on the state:

**Lemma 2** (Itô's lemma [Särkkä and Solin, 2019]). *If $W_t \in \mathbb{R}^d$ is a Brownian motion on $[0, T]$, and $f(x) : \mathbb{R}^d \to \mathbb{R}$ is a twice continuously differentiable function, then for any $t \leq T$,*

$$df(W_t) = \frac{1}{2}\mathrm{tr}(\mathrm{Hess}_x f(W_t))dt + \nabla_x f(W_t)^T dW_t. \quad (5)$$

With these tools in mind, we describe how to efficiently calculate solutions of parabolic PDEs with changing drift and volatility functions.

# 3 LEARNING A BASE MODEL

Recall that we want to develop a model with the following properties such that it applies to many tasks:

- Adaptability, the model defines a representation shared across many tasks;
- Extensibility, the representation can be rapidly assimilated to new tasks.

To achieve these properties, we invoke Definition 1 and Lemma 2 to a base set of sample paths shared across all tasks. We first describe the general approach and why naively applying Definition 1 fails in many cases, which motivates the need for neural parameterization. We then consider maximizing a function of parabolic PDE solutions rather than solving the PDEs. We finally present the operator-learning framework where different tasks correspond to the solutions to a PDE under different parameter configurations.

---

[2]To avoid confusion with diffusion processes, we will refer to $\sigma$ as the *volatility*, but note that it often denotes the diffusion coefficient.

## 3.1 SOLVING PARABOLIC PDES THROUGH IMPORTANCE SAMPLING

Consider a parabolic PDE that satisfies (1) using the representation given in (2). Computing the solution for (1) for a series of $K$ different drifts, $\{\mu^{(i)}\}_{i=1}^K$, requires simulating a different SDE for each $\mu^{(i)}$. Simulating the SDEs with an Euler-Maruyama integration, where the discretization size is $h$, the number of time steps is $N_T = T/h$, and approximating the expectation with $N_E$ different realizations requires $N_E \times N_T \times K$ computations. Importantly, to compute $X_t$, the computation *must be performed sequentially*, since it relies on the previous value of $X_t$. This sequential operation induces the main bottleneck, and we try to avoid it whenever possible.

We instead consider the SDE $dY_t = \sigma(Y_t)dW_t$ associated with marginal distribution $\mathbb{P}_{Y_t}$ which we assume we can simulate as a function of Brownian motion $f(W_t)$. Suppose we wish to solve the PDE in (1) for $\mu = \{\mu^{(i)}(x)\}_{i=1}^K, \sigma = \sigma(x)$. Using (4) combined with (2), we can write the solution with $\mu^{(i)}$ write the expectation as

$$p_{\mu^{(i)}}(T, x) :=$$

$$\mathbb{E}_{\mathbb{P}_{X_t^{(i)}}}\left[p_0(X_T^{(i)})\exp\left(-\int_0^T r\left(X_s^{(i)}\right)ds\right) \mid X_0^{(i)} = x\right]$$

$$= \mathbb{E}_{\mathbb{P}_{Y_t}}\left[p_0(Y_T)\exp\left(-\int_0^T r(Y_s)ds\right)\frac{d\mathbb{P}_{X_T^{(i)}}}{d\mathbb{P}_{Y_T}} \mid Y_0 = x\right]$$

$$(6)$$

where $X_t$ satisfies $dX_t^{(i)} = \mu^{(i)}(X_t^{(i)})dt + \sigma(X_t^{(i)})dW_t$ with measure $\mathbb{P}_{X_t^{(i)}}$. By writing the expectation with respect to $Y_t$, we can reuse the $N$ simulations of $Y_t$ for each $\mu^{(i)}$, and we only need to compute the likelihood ratio, $\frac{d\mathbb{P}_{X_T^{(i)}}}{d\mathbb{P}_{Y_T}}$ for $i = 1 \ldots K$. This formulation is crucial since each $\frac{d\mathbb{P}_{X_T^{(i)}}}{d\mathbb{P}_{Y_T}}$ only requires an integral (approximated by a sum) rather than sampling an SDE. Unfortunately upon inspecting (3), computing the likelihood ratio $\frac{d\mathbb{P}_{X_T}}{d\mathbb{P}_{Y_T}}$ requires computing the exponential of numerically approximated integrals. This results in an error that grows exponentially in the discretization size $h$ rather than linearly in $h$, as is the case when directly computing the expectation with new sample paths from each $\mu^{(i)}$. In the next section, we discuss ways to circumvent this issue by deriving a lower bound of the PDE solution with an error linear in $h$ and developing a neural network-based representation of the likelihood ratio.

## 3.2 MAXIMIZING PARABOLIC PDES

It is often the case that we are interested in the parameters that *maximize* the expectation of a function of sample paths, i.e. $\max \mathbb{E}\left[J\left(X_T^{(i)}\right)\right]$ where the maximization is

over parameters of $X_T$ for different tasks associated with $K$ distinct datasets, $\{X^{(i)}\}_{i=1}^K$. Note that by Lemma 2, this expectation satisfies the PDE (1). To motivate this approach, we will consider the problem of sampling from a family of target distributions, as illustrated on the left side of Figure 1. This is closely related to the works on diffusion models in [Song et al., 2021a,b],which uses a neural network to approximate the score function of a transition density described by an SDE. However, in those cases, a specific form of the forward SDE is used such that an analytical form of the transition density exists. Instead, we are interested in relaxing this parametric assumption and working directly with the Fokker-Planck equation through its stochastic representation.

Continuing with $K$ target distributions $\{p^{(i)}(x)\}_{i=1}^K$ that we wish to approximate, when approximating with an Itô diffusion, we can represent each distribution as the solution of the Fokker-Planck equation at some terminal time $T$ under $K$ different parameters. This equation is a linear parabolic PDE which we can write in terms of an expectation according to the Feynman-Kac formula (2) with $r = \nabla \cdot \mu^{(i)}$ (see Appendix G for derivation) and $dX_t^{(i)} = \mu^{(i)}(t, X_t^{(i)})dt + \sigma(t)dW_t$, assuming that $\sigma$ is independent of $X_t$ for ease of explanation. Then, we want to maximize the distribution's likelihoods for $K$ different tasks at $T$. We will use the idea described in Section 3.1 to bypass the expensive Euler-Maruyama sampling procedure during each training iteration. To do this, we use the form in (6) with the parameters of the latent process $Y_t$ being the meta-learned parameters, i.e. $dY_t = \mu_0(Y_t, t)dt + \sigma_0(t)dW_t$, $Y_0 \sim p_0$. As such, we have translated the problem from requiring $X_t$ to $Y_t$ samples, which are reusable across training iterations. We take the parameters $(p_0, \mu_0, \sigma_0)$ as the *meta parameters*, which are optimized over all $K$ tasks.

The exponential term in (4) still incurs high errors for large $T$, and solving the PDE using this form is inaccurate without prohibitively large $N_h$. To circumvent this, we remind ourselves that the goal is to *maximize* the solution of the PDE, which corresponds to the likelihood in this case. Applying Jensen's inequality, we obtain an evidence lower bound (ELBO) without the exponential error:

$$\log p(T, x) \geq \mathbb{E}_{\mathbb{P}_{X_t}} \left[ \int_0^T \nabla \cdot \mu(s, X_s)ds + \log p_0(X_T) \right]$$
$$(7)$$

$$= \mathbb{E}_{\mathbb{P}_{Y_t}} \left[ \int_0^T \hat{\mu}(s, Y_s)dW_s - \int_0^T \frac{1}{2}\hat{\mu}^2(s, Y_s) \right.$$
$$\left. - \nabla \cdot \mu(s, Y_s)ds + \log p_0(Y_T) \mid Y_0 = x \right]$$
$$(8)$$

with (8) denoted as $\text{ELBO}_{\text{IS}}$, with IS referring to *importance sampling*, and (7) denoted as $\text{ELBO}_{\text{direct}}$ and $\hat{\mu} = (\sigma_0\sigma_0^\top)^{-1}\mu, \hat{\mu}^2 = \mu^\top(\sigma_0\sigma_0^\top)^{-1}\mu$. Huang et al. [2021]

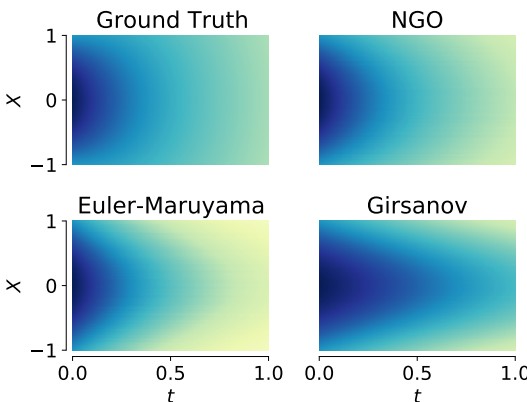

Figure 2: Simulated solutions of the Fokker-Planck equation of an $1d$ OU-process compared with the analytic solution.

explored a similar idea for score-based diffusion models and proved that the bound in (7) is tight when maximizing over a sufficiently expressive class of drift parameterizations. Circling back to the maximum likelihood estimation problem, we can apply this technique to reduce the exponential error to linear while reusing the sample paths $Y_t$ and meta-learning the parameters $(p_0, \mu_0, \sigma_0)$. We describe an explicit example of meta-learning $p_0$ in Appendix G.

### 3.3 NGO: META-LEARNING A CONTINUOUS SPACE OF TASKS

Having described the meta-learning framework where we consider $(p_0, \mu_0, \sigma_0)$ as meta-learned parameters for the case of maximizing the solution to a parabolic PDE, we now study estimating the explicit *solution* to the PDE. Specifically, we want a model that easily generalizes to solutions for different parameters in one shot. From the previous section, we can apply importance sampling to compute the solution for different $\mu^{(i)}$'s while reusing the sample paths. However, we noted that this incurs an exponential error for the integration in time when computing with the Euler-Maruyama method. The key idea of the operator learning approach is to instead learn an optimal integrator through a neural network for a parametric family of drifts. We refer to this as the Neural Girsanov Operator (NGO), which transforms the expected value of solutions of SDEs $Y_t$ with meta-parameters $(p_0, \mu_0, \sigma_0)$ to SDEs $X_t$ with drift $\mu^{(i)}$. We will consider the case where $\mu_0 = 0, \sigma_0 = 1$ for ease of exposition. This approximation of $\mathbb{E}[p_0(X_T) \mid \mathcal{F}_T]$ is done by equating the following:

$$\mathbb{E}_{\mathbb{P}_X}[p_0(X_T) \mid \mathcal{F}_T] \approx$$
$$\mathbb{E}_{\mathbb{P}_Y}\left[ p_0(Y_t)\text{NGO}\left(\{\mu(Y_{s_n}), \Delta W_{s_n}, h\}_{n_T=1}^{N_T}; \theta\right) \mid \mathcal{F}_T \right]$$

where $\text{NGO}(\cdot; \theta)$ is a neural network with non-negative outputs with parameters $\theta$. For a parametric set of drifts

$\{\mu^{(i)}(\cdot; \xi_i)\}$, the NGO learns the optimal numerical integrator over the parameter space. We parameterize NGO using a convolutional neural network, motivated by the connection between finite different stencils and convolutions, although the integration performed by NGO involves nonlinear terms. To optimize the parameters of NGO, we consider the following optimization over the measure $\nu(\xi)$ of all parameterizations of drifts $\mu$ we are interested in solving:

$$\min_{\theta} \mathbb{E}_{\mu_\xi} \Bigg( \mathbb{E}\left[ p_0(Y_T) \text{NGO}(\{\mu_\xi(Y_{s_n}), \Delta W_{s_n}, h\}_{n_T=1}^{N_T}; \right.$$
$$\left. \theta) \right] - \mathbb{E}\left[ p_0(Y_T) \frac{d\mathbb{P}_{X_{\mu_\xi}}}{d\mathbb{P}_Y} \right] \Bigg)^2. \tag{9}$$

The loss in (9) has a few helpful properties. First, this loss only requires sampling Brownian motion and approximating the integral in (3). This property is a notable departure from the usual requirement of the exact solution and parameter values — we do not need to solve for the explicit solution of the PDE using Euler-Maruyama but only need to sample Brownian motion and approximate an integral. Second, although the second term incurs high numerical error, since we average over many realizations, this error does not affect the approximation of the solution. Finally, this solution does not require a meshing of the domain and provides solutions depending on the starting points of the Brownian motion, which can be arbitrary. This attribute is vital for evaluating solutions over complex domains. Algorithm 1 describes the full algorithm.

---

**Algorithm 1** Approximating linear parabolic PDEs with NGO

---

**Input:** $N \in \mathbb{N}$, $h \in \mathbb{R}_+$, $\mu(t, x): \mathbb{R}_+ \times \mathbb{R}^d \to \mathbb{R}^d$, evaluation coordinate $(T, X)$

1: Sample $N$ Brownian motions to time $T$ starting at $X$, $\left\{ X + \sqrt{kh} \varepsilon^{(i)} \right\}_{k=1...T/h}^{i=1...N}$, $\varepsilon \sim \mathcal{N}(0, 1)$

2: **for** $i \in \{1, \ldots, N\}$ **do** ▷ Easy to parallelize.

3:      Compute $\frac{d\mathbb{P}_\mu^{(i)}}{d\mathbb{P}_W} \approx$ $\text{NGO}\left[ \left\{ \mu\left( W_k^{(i)} \right) \right\}_{k=1}^{T/h}, \left\{ \sqrt{kh} \varepsilon^{(i)} \right\}_{k=1}^{T/h}, h \right]$ ▷ Stochastic exponential.

4: **end for**

**Output:** Approximation of $u(T, X)$ as $\check{u}(T, X) = \frac{1}{N} \sum_{i=1}^{N} p_0(W_T^{(i)}) \frac{d\mathbb{P}_\mu^{(i)}}{d\mathbb{P}_W}$

---

**Extension to semi-linear parabolic PDEs**    We presented the method in terms of linear parabolic PDEs. However, extending to semi-linear parabolic PDEs is relatively straightforward by again using the stochastic representation of such

PDEs. We can consider equations of the form:

$$\begin{cases} \frac{\partial p}{\partial t} + \frac{1}{2}\text{Tr}(\sigma\sigma^\top(t, x)\text{Hess}_x p(t, x)) + \nabla_x p(t, x)^\top \mu(t, x) \\ \quad + \phi(x, p, \sigma^\top \nabla p) = 0, \\ p(T, x) = g(x). \end{cases}$$
$$\tag{10}$$

This PDE has a stochastic representation:

$$dX_t = \mu(t, x)dt + \sigma(t, x)dW_t, \qquad X_0 = x;$$
$$dS_t = -\phi(x, p, \sigma^\top \nabla_x p)dt + Z_t^\top dW_t, \quad S_T = g(X_T),$$

and $p(t, x) = \mathbb{E}[S_t \mid X_0 = x]$ [Exarchos and Theodorou, 2018, Bender and Moseler, 2010]. Simulating this system requires computing two Euler schemes: one for the forward component $X_t$ and the other for the backward process $S_t$. We can easily follow the scheme for the forward component $X_t$ presented in Algorithm 1 by sampling Brownian motion and computing an expectation with the estimated exponential martingale. If $\phi$ does not depend on $p$, we can compute the integration of $S_t$ using a basic sum without requiring sequential computations. We provide the complete algorithm in Appendix E.

**Extension to elliptic PDEs**    Finally, we note that extending to elliptic PDEs is also relatively straightforward. Elliptic PDEs require computing the first hitting time of the domain boundary $\partial \mathcal{D}$ at each evaluation point. Then, using the first hitting times, the importance sampling follows. Specifically, we modify the stochastic exponential in (3) to be: $\frac{d\mathbb{P}_Y}{d\mathbb{P}_X} := \exp\left(\int_0^\tau \mu(X_s)dW_s - \frac{1}{2}\int_0^\tau \mu(X_s)^2 dt\right)$ where $\tau$ is the first hitting time of $\partial \mathcal{D}$ starting at $x$.

## 4   PROPERTIES OF THE ESTIMATORS

Since the meta-learning framework follows from the stochastic representation of this class of PDEs, theoretical analysis is particularly amenable in contrast to other black-box methods that only use neural networks. We discuss the error rate of the ELBO in (8) and the convergence rate of the NGO over a parametric space of solutions.

### 4.1   ERROR ANALYSIS

We first note that the proposed algorithm induces a trade-off between memory and execution time since we save the Brownian motions underlying the importance sampling. Saving the Brownian motions is a minor constraint since the Brownian motions saved are only $\mathcal{O}(N_E \times d)$ where $N_E$ is the number of Monte Carlo samples used to estimate the expectation, and $d$ is the number of dimensions. Additionally, these can be distributed over multiple devices, as no communication between nodes is needed when computing the expectation. We also analyze the approximation error of both $\text{ELBO}_{\text{IS}}$ and $\text{ELBO}_{\text{direct}}$ presented in Section 3.2.

Although ELBO$_{\text{IS}}$ introduces more errors than ELBO$_{\text{direct}}$ by having more integration terms, they are all of at least order $\mathcal{O}(h^2)$. Employing a multi-level architecture based on the multi-level Monte Carlo can improve the accuracy further under a similar computational budget [Giles, 2015].

## 4.2 UNIFORM CONVERGENCE OVER DRIFT PARAMETERS

A final property of the NGO concerns the convergence rate over a family of solutions to PDEs with parameter $\mu(x,t)$ and $\sigma(t)$ dependent only on $t$. Specifically, by using the properties of the stochastic representation, we can show that a well-learned NGO-based solution $p_\theta^\xi(x)$ uniformly converges over the parameter space $\Xi$ to the ground truth $p^\xi(x)$ under mild conditions. Intuitively, since the NGO learns how to compute the likelihood ratio, we can change the parameters within a compact set while maintaining high performance over this set.

**Proposition 1** (Uniform Convergence)**.** *For fixed $x \in \mathcal{D}, T \in \mathbb{R}_+$, consider a space of functions $\mathcal{F} = \left\{ d\mathbb{P}_{X_T^{(\xi)}}/d\mathbb{P}_{Y_T} : \xi \in \Xi \right\}$ parameterized by $\xi$ from a compact set $\Xi \subset \mathbb{R}^k$ satisfying $Var\left( d\mathbb{P}_{X_T^{(\xi)}}/d\mathbb{P}_{Y_T} \right) < \infty$ for all $\xi \in \Xi$ with $\mathbb{P}_{X_T^{(\xi)}}$ denoting the distribution of the solution $X_T = x + \int_0^T \mu(X_t, t; \xi)\mathrm{d}t + \int_0^T \sigma(t)\mathrm{d}W_t$ and $\mathbb{P}_{Y_T}$ the distribution of $Y_t = x + \int_0^T \sigma(t)\mathrm{d}W_t$. Additionally, assume that the image of $(T, X_T) \mapsto \mu(T, X_T; \xi)$ is compact for all $X_T, \xi$. Then, $\mathbb{G}_{N_E} = \sqrt{N_E}\left( p_\theta^\xi(x, T) - p^\xi(x, T) \right)$ converges in distribution to a zero-mean Gaussian process over $\xi \in \Xi$ as $N_E \to \infty$ where $N_E$ is the number of samples used to compute the expectation.*

The proof follows from first showing that $\mathcal{F}$ is $\mathbb{P}_{Y_T}$-Donsker and then follows with an analysis of our construction of the solution $p_\theta^\xi$ in terms of an expectation. The complete statement is in Appendix A.1.

## 5 EXPERIMENTS

We now examine the capabilities of the models in their respective tasks. First, we illustrate a proof-of-concept experiment on maximizing the parameters of a PDE by estimating $K$ target distributions in a generative modeling setting. Then, we present our main experiments on solving parabolic PDEs. We simulate the sample paths with the basic Euler-Maruyama solver for all experiments.

### 5.1 MAXIMIZING PARAMETERS

To illustrate the running example on generative modeling, we consider maximizing the solution of the Fokker-Planck

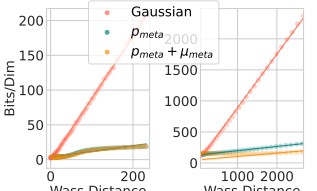 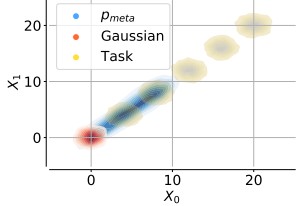

(a) Sampling $2d$ (left) and $100d$ (right) Gaussians with different means starting from a standard Gaussian versus $p_{\text{meta}}$ and no meta-drift versus $\mu_{\text{meta}}$.

(b) Comparing $p_{\text{meta}}$, a standard Gaussian, and the target task distributions. $p_{\text{meta}}$ provides the best initial condition for all the target distributions.

Figure 3: Numerical results for meta-learning generative models for Gaussian distributions where different means correspond to different tasks.

PDE (corresponding to the likelihood of a generative model) at a terminal time using (8). Consider the problem of approximating $K$ Gaussian target distributions with different means and the same covariance matrix. We are interested in investigating the sample quality in the few-shot learning setting with and without the meta-learned parameter by training $K$ separate drift and diffusion functions $\{\mu_i, \sigma_i\}_{i=1}^K$ on $K$ different target distributions. We represent the meta-parameter $p_{\text{meta}}$ as a small normalizing flow, which we optimize over $\kappa < K$ training distributions similar to the $K$ target distributions. To sample the $i^{\text{th}}$ target distribution parameterized by $\mu_i$ and $\sigma_i$, we first sample from the initial distribution (either a standard Gaussian or the meta-learned $p_{\text{meta}}$) and evolve the SDE to the terminal time according to $\mu_i$ and $\sigma_i$. Figure 3a shows the test distribution bits/dim for the 100 target distributions compared to the 2-Wasserstein distance between the initial and target distributions. Figure 3b visualizes $p_{\text{meta}}$, standard Gaussian, and sampled target distributions in the $2d$ case. The results demonstrate the importance of including the meta-learned parameter in the optimization to improve generalization.

### 5.2 OPERATOR LEARNING

We consider examples of the NGO on operator learning tasks by testing on a few PDEs. We first consider a linear and semi-linear PDE, with parametric classes of the function $\mu$ in the linear PDE (1) and $\mu$, $h$ in the semi-linear PDE (10). Figure 2 in the Appendix shows a visualization of solutions of a $1d$ Fokker-Planck equation calculated with the analytical solution, NGO, Euler-Maruyama, and directly with Girsanov. We then consider canonical parabolic PDEs consisting of two linear equations – the Black-Scholes (BS) and the Fokker-Planck (FP) equations, and two semi-linear equations – the Hamilton-Jacobi-Bellman (HJB) and the Black-Scholes-Barrenblatt (BSB) equations. These equations have applications in finance (BS, BSB), stochastic control (HJB), and the previously explored probabilistic modeling (FP).

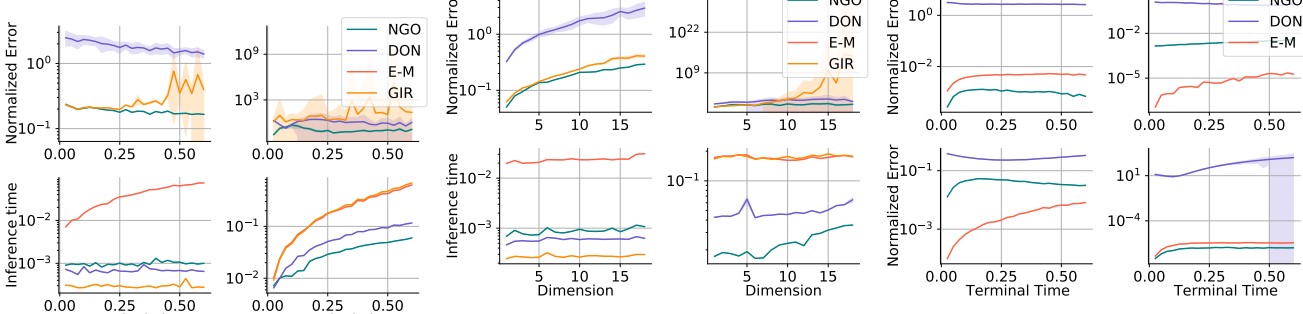

(a) $10d$ linear (left) and semi-linear (right) parabolic PDEs.

(b) Varying the dimensionality at $T = 0.25$ for linear (left) and semi-linear (right) parabolic PDEs.

(c) Normalized errors for $10d$ HJB (top left), BSB (top right), BS (bottom left), and FP (bottom right).

Figure 4: Comparison of the normalized errors and inference times of NGO, DeepONet (DON), Girsanov (GIR), and Euler-Maruyama (E-M) on linear and semi-linear parabolic PDEs.

We present detailed definitions of the PDEs in Appendix D. For this study, we compare against the DeepONet operator learning architecture [Lu et al., 2019] (DON) with a similar model size, naively applying the change-of-measure (GIR), and the direct simulation with Euler-Maruyama (E-M). Note that the E-M method provides a strong baseline that encompasses many techniques in the related work (e.g. [Berner et al., 2020, Glau and Wunderlich, 2022, Richter and Berner, 2022]), so we use this as a baseline for the existing deep learning methods based on Feynman-Kac. We compare computation time and accuracy between the different methods for estimating the solution under different $\mu$ and $h$ functions. When analytical solutions do not exist, we consider E-M with a substantial $N_E, N_T$ as the ground truth.

**PDEs with defined basis**    In the linear case, we consider second-order polynomials

$$\mu^{(i)}(x) \in \left\{ \sum_{i=0}^{2} c_i x^i \mid c_i \sim \mathcal{U}(0,1) \right\}.$$

For the semi-linear equation, we also test changing the backward drift $\phi$ in (10) by considering basis functions given by

$$\phi_i(t, x, s, z) \in \{c_1 \sum \sin(x_i) + c_2 \sum z_i^2 + c_3 \cos(t+s) \\ \mid c_i \sim \mathcal{U}(0,1)\}.$$

We set the parameters $r = 0$ and $\sigma = 1$ for these experiments. We randomly sample from these function classes during training and then evaluate on a different test set of functions. The results are illustrated in Figure 4 and Table 1 for these two equations with Figure 4a presenting the error and computation at different terminal times and Figure 4b considering the error and inference time at various dimensions. The proposed NGO has high accuracies while maintaining small computation times in all the tested regimes.

**Canonical parabolic PDEs**    We test the generalization capabilities of NGO and DON models trained in the previous section on four canonical parabolic PDEs previously mentioned (BS, FP, HJB, and BSB) in $10d$. Since an exact solution is known for these equations, we compare NGO, DeepONet, and E-M to the analytical solution (presented in Appendix D). Note that for the BS and BSB equations, a change in the volatility function $\sigma$ occurs. Results on the normalized error are in Figure 4c. The proposed NGO achieves low errors across all four tested canonical PDEs again. Additional ablations are available in Appendix C.

| | # Param. | Loss | Inf. Time (s) |
|---|---|---|---|
| NGO | 15.7K | $4.50(0.05) \times 10^{-2}$ | $2.7 \times 10^{-2}$ |
| FNO | 19.9K | $4.44(0.16) \times 10^{-2}$ | $7.4 \times 10^{-2}$ |
| NGO | 18.8K | $5.10(0.11) \times 10^{-2}$ | $1.80 \times 10^{-2}$ |
| FNO | 2.0M | $6.10(0.02) \times 10^{-2}$ | $1.14 \times 10^{-1}$ |

Table 1: NGO and Fourier Neural Operator (FNO) performances on one-dimensional (first two rows) and two-dimensional (last two rows) linear parabolic equations. We calculate the normalized losses and standard errors with five independent training episodes.

## 6    DISCUSSION

We proposed a method for solving problems related to parabolic PDEs based on their stochastic representation. We treat the parameters of parabolic PDEs' stochastic representation as the meta-learned parameters shared across all tasks and calculate task-specific solutions with them. This structure allows application in optimizations under different scenarios and solving PDEs with distinct parameters through the NGO. Empirical results indicate that NGO provides a sizable advantage in computation time and accuracy compared to baselines.

**Limitations** Theoretically, if the target drift has a large magnitude, the variance of stochastic exponential can be high, which may lead to numerical instabilities. In this case, the direct Euler-Maruyama approach may be beneficial for training the neural operator.

## Acknowledgements

This work was supported in part by the Office of Naval Research (ONR) under grant number N00014-21-1-2590. AH was supported by NSF-GRFP.

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

# A ADDITIONAL SIMULATION RESULTS

**Proposition 2.** *Approximating the* ELBO$_{IS}$ *and* ELBO$_{direct}$ *terms with Euler-Maruyama using step size h will both induce the error* $h \int_0^t \mathbb{E}[\psi_e(s, X_s)\mathrm{d}s] + \mathcal{O}(h^2)$, *where*

$$\psi_e(t,x) = \frac{1}{2}\sum_{i,j=1}^d \mu^i(t,x)\mu^j(t,x)\partial_{x_i x_j}p(t,x) + \frac{1}{2}\sum_{i,j,k=1}^d \mu^i(t,x)a_k^j(t,x)\partial_{x_i x_j x_k}p(t,k)$$

$$+ \frac{1}{8}\sum_{i,j,k,l=1}^d a_j^i(t,x)a_l^k(t,x)\partial_{x_i x_j x_k x_l}p(t,x) + \frac{1}{2}\frac{\partial^2}{\partial t^2}p(t,x)$$

$$+ \sum_{i=1}^d \mu^i(t,x)\frac{\partial}{\partial t}\partial_{x_i}u(t,x) + \frac{1}{2}\sum_{i,j=1}^d a_j^i(t,x)\frac{\partial}{\partial t}\partial_{x_i x_j}u(t,x)$$

*and* $a(t,x) = \sigma(t,x)\sigma^\top(t,x)$.

*Proof.* For this section, we do not consider Monte Carlo error $N_E$ and focus only on the integration error. We analyze the approximation error of ELBO$_{direct}$ and ELBO$_{IS}$ using the error bound introduced in [Talay and Tubaro, 1990]. Given the following SDE:

$$\mathrm{d}X_t = \mu(X_t, t)\mathrm{d}t + \sigma(X_t, t)\mathrm{d}W_t$$

We want to estimate the error of $\mathbb{E}[f(X_T)]$, where we evaluate $X_T$ through the Euler-Maruyama scheme. Define $h = T/N$ as the step size in the Euler-Maruyama scheme and denote $X_T^h$ as the approximated $X_T$ using step size $h$. Following Talay and Tubaro [1990], we define the error term $err(T, h) = \mathbb{E}[f(X_T)] - \mathbb{E}[f(X_T^h)]$.

Talay and Tubaro [1990] proved that

$$err(T, h) = -h \int_0^T \mathbb{E}[\psi_e(s, X_s)]\mathrm{d}s + \mathcal{O}(h^2) \tag{11}$$

where

$$\psi_e(t,x) = \frac{1}{2}\sum_{i,j=1}^d \mu^i(t,x)\mu^j(t,x)\partial_{x_i x_j}u(t,x)$$

$$+ \frac{1}{2}\sum_{i,j,k=1}^d \mu^i(t,x)a_k^j(t,x)\partial_{x_i x_j x_k}u(t,k)$$

$$+ \frac{1}{8}\sum_{i,j,k,l=1}^d a_j^i(t,x)a_l^k(t,x)\partial_{x_i x_j x_k x_l}u(t,x) + \frac{1}{2}\frac{\partial^2}{\partial t^2}u(t,x)$$

$$+ \sum_{i=1}^d \mu^i(t,x)\frac{\partial}{\partial t}\partial_{x_i}u(t,x) + \frac{1}{2}\sum_{i,j=1}^d a_j^i(t,x)\frac{\partial}{\partial t}\partial_{x_i x_j}u(t,x)$$

and $a(t,x) = \sigma(t,x)\sigma^T(t,x)$.

We will assume that $\sigma = 1$ for ease of analysis. For the ELBO, we have an additional integral we must approximate related to the divergence of the drift term.

Recall that

$$\text{ELBO}_{direct} = \mathbb{E}_{\mathbb{P}_{X_t}}\left[\int_0^T \nabla \cdot \mu(s, X_s)\mathrm{d}s + \log p_0(X_T) \mid X_0 = x\right]$$

$$\text{ELBO}_{IS} = \mathbb{E}_{\mathbb{P}_{Y_t}}\left[\int_0^T \mu(s, Y_s)\mathrm{d}W_s - \int_0^T \frac{1}{2}\mu(s, Y_s)^T \mu(s, Y_s) - \nabla \cdot \mu(s, Y_s)\mathrm{d}s + \log p_0(Y_T) \mid Y_0 = x\right]$$

The error associated with the divergence is defined as

$$err_{\text{div},t} = -h \int_0^t \mathbb{E}[\psi_e(s, X_s)]\mathrm{d}s + \mathcal{O}(h^2)$$

with a similar argument following the initial condition, i.e.

$$err_{p_0} = -h \int_0^T \mathbb{E}[\psi_e(s, X_s)]\mathrm{d}s + \mathcal{O}(h^2)$$

Combining these terms, we get the final error, $err_{\text{ELBO}_{\text{direct}}}$:

$$
\begin{aligned}
err_{\text{ELBO}_{\text{direct}}} &= \int_0^T err_{\text{div},s}\mathrm{d}s + err_{p_0} \\
&= \sum_{i=1}^N h \times err_{\text{div},t_i} + err_{p_0} \\
&= N \times h[-h \int_0^T \mathbb{E}[\psi_e(s, X_s)] + \mathcal{O}(h^2)] - h \int_0^T \mathbb{E}[\psi_e(s, X_s)]\mathrm{d}s + \mathcal{O}(h^2) \\
&\approx -h \int_0^T \mathbb{E}[\psi_e(s, X_s)] + \mathcal{O}(h^2).
\end{aligned}
$$

For the $\text{ELBO}_{\text{IS}}$ case, $X_t$ can be sampled exactly since we assume it follows a Brownian motion, which removes the integration error in $err_{p_0}$.

The errors are then introduced when integrating the terms in the stochastic exponential. Then we have

$$\underbrace{\int_0^T \nabla \cdot \mu(s, Y_s)\mathrm{d}s}_{err_{\text{div}}} - \underbrace{\int_0^T \frac{1}{2}\mu(s, Y_s)^T \mu(s, Y_s)\mathrm{d}s + \int_0^T \mu(s, Y_s)\mathrm{d}W_s}_{\mathcal{O}(h)}$$

where the second error rate comes from the Euler discretization. Following this argument, we get

$$err_{\text{ELBO}_{\text{IS}}} = -h \int_0^T \mathbb{E}[\psi_e(s, X_s)]\mathrm{d}s + \mathcal{O}(h^2). \tag{12}$$

Although $\text{ELBO}_{\text{IS}}$ introduces more error than $\text{ELBO}_{\text{direct}}$ by having more integration terms, they are all of the order $\mathcal{O}(h^2)$. $\qquad\square$

## A.1 UNIFORM CONVERGENCE

**Proposition 3** (Uniform Convergence). *For fixed $x \in \mathcal{D}, T \in \mathbb{R}_+$, consider a space of functions*

$$\mathcal{F} = \left\{ \mathrm{d}\mathbb{P}_{X_T^{(\xi)}}/\mathrm{d}\mathbb{P}_{Y_T} : \xi \in \Xi \right\}$$

*parameterized by $\xi$ from a compact set $\Xi \subset \mathbb{R}^k$ satisfying $Var\left(\mathrm{d}\mathbb{P}_{X_T^{(\xi)}}/\mathrm{d}\mathbb{P}_{Y_T}\right) < \infty$ for all $\xi \in \Xi$ with $\mathbb{P}_{X_T^{(\xi)}}$ denoting the distribution of the solution $X_T = x + \int_0^T \mu(X_t, t; \xi)\mathrm{d}t + \int_0^T \sigma(t)\mathrm{d}W_t$ and $\mathbb{P}_{Y_T}$ the distribution of $Y_t = x + \int_0^T \sigma(t)\mathrm{d}W_t$. Additionally, assume that the image of $(T, X_T) \mapsto \mu(T, X_T; \xi)$ is compact for all $X_T, \xi$. Then,*

$$\mathbb{G}_{N_E} = \sqrt{N_E}\left(p_\theta^\xi(x, T) - p^\xi(x, T)\right)$$

*converges in distribution to a zero-mean Gaussian process over $\xi \in \Xi$ as $N_E \to \infty$ where $N_E$ is the number of samples used to compute the expectation.*

*Proof.* First, we will assume that the stochastic exponential has a finite variance for all $t, x$ within the support of the distribution. The finite variance allows us to use the law of large numbers to obtain pointwise convergence of the empirical expectation to the ground-truth solution. Additionally, we assume that the operator NGO is well-learned in the sense that $\text{NGO}(x, T, \xi) = \frac{d\mathbb{P}_{X_T^{(\xi)}}}{d\mathbb{P}_{Y_T}}$ can be computed exactly.

Next, we need to show that the class of functions we are approximating are $P$-Donsker, which we will do using a covering number argument. Recall that the parameters of the functions are assumed to be from a compact set $\Omega \subset \mathbb{R}^d$ with

$$\mathcal{F} = \left\{ f(T, \{X_t\}; \xi) := \exp\left( \int_0^T \mu(X_t; \xi) dW_t - \frac{1}{2} \int_0^T \mu(X_t; \xi)^2 dt \right) : \xi \in \Xi \right\}$$

also compact. The covering number of $\Xi$, a subset of the Euclidean space, is known to be bounded by $N(\varepsilon, \Xi, \|\cdot\|) \leq C \left( \frac{1}{\varepsilon} \right)^d$ for some $C > 0$. We will use this to bound the bracketing number of $\mathcal{F}$. The $\log$ of the covering number is then bounded by $O(d \log \frac{1}{\varepsilon}) < \frac{1}{\varepsilon^2}$.

Using this to bound the bracketing number, we can obtain that $\mathcal{F}$ is $P$-Donsker (c.f. Sen [2018, Theorem 11]). We can then define the empirical process $\mathbb{G}_{N_E}$, $N_E$ corresponding to the number of terms used to take the expectation as

$$\mathbb{G}_{N_E}^{\xi} := \sqrt{N_E} \left( \mathbb{P}_{Y_T}^{N_E} - \mathbb{P}_{Y_T} \right) p_0(Y_T) \text{NGO}(\cdot; \xi)$$

$$= \sqrt{N_E} \left( \sum_{i=1}^{N_E} p_0(Y_T^{(i)}) \text{NGO}(Y_T^{(i)}; \xi) - \mathbb{E}\left[ p_0(Y_T) \text{NGO}(Y_T; \xi) \right] \right).$$

Since NGO is assumed to be well learned, it approximates the likelihood ratio exactly, so the expectation gives the ground-truth solution. From the finite variance assumption on NGO, by the central limit theorem for any $\xi \in \Xi$, $\mathbb{G}_{N_E}^{\xi} \to \sqrt{N_E}\mathcal{N}(0, 1)$. Given that the finite-dimensional margins are unit normal, we conclude that $\mathbb{G}$ converges to a Gaussian process over $\Xi$. $\square$

# B  NETWORK STRUCTURE AND TRAINING HYPERPARAMETERS

## B.1  LINEAR PARABOLIC PDE

**Network structure**  We first provide network structures on the NGO and DeepONet used to calculate solutions of linear parabolic PDEs. For both NGO and DeepONet, we use a network structure that correlates to the number of dimensions of the PDEs, as shown in Tables 2,3,4.

| Operation Layer | Input Channels | Output Channels |
|---|---|---|
| Convolutional Layer | $N_{\dim} \times 2 + 1$ | $N_{\dim} \times 5 + 50$ |
| Softplus | N/A | N/A |
| Convolutional Layer | $N_{\dim} \times 5 + 50$ | $N_{\dim} \times 5 + 50$ |
| Softplus | N/A | N/A |
| Convolutional Layer | $N_{\dim} \times 5 + 50$ | $N_{\dim} \times 5 + 50$ |
| Softplus | N/A | N/A |
| Convolutional Layer | $N_{\dim} \times 5 + 50$ | $N_{\dim} \times 5 + 50$ |
| Softplus | N/A | N/A |
| Convolutional Layer | $N_{\dim} \times 5 + 50$ | $N_{\dim} \times 5 + 50$ |
| Softplus | N/A | N/A |
| Convolutional Layer | $N_{\dim} \times 5 + 50$ | $N_{\dim} \times 5 + 50$ |
| Softplus | N/A | N/A |
| Convolutional Layer | $N_{\dim} \times 5 + 50$ | $N_{\dim}$ |

Table 2: Network structure of NGO in the linear parabolic case.

| Operation Layer | Input Number | Output Number |
|---|---|---|
| Linear Layer | $N_{\text{sensor}} = 100$ | $N_{\text{dim}} \times 5 + 70$ |
| Tanh | N/A | N/A |
| Linear Layer | $N_{\text{dim}} \times 5 + 70$ | $N_{\text{dim}} \times 5 + 70$ |
| Tanh | N/A | N/A |
| Linear Layer | $N_{\text{dim}} \times 5 + 70$ | $N_{\text{dim}} \times 5 + 70$ |
| Tanh | N/A | N/A |
| Linear Layer | $N_{\text{dim}} \times 5 + 70$ | $N_{\text{dim}} \times 5 + 70$ |
| Tanh | N/A | N/A |
| Linear Layer | $N_{\text{dim}} \times 5 + 70$ | $N_{\text{branch}} = 15$ |

Table 3: Network structure of the "branch" network of the DeepONet in both the linear and semi-linear parabolic caseGoswami et al. [2022].

| Operation Layer | Input Number | Output Number |
|---|---|---|
| Linear Layer | $N_{\text{dim}} + 1$ | $N_{\text{dim}} \times 5 + 50$ |
| Tanh | N/A | N/A |
| Linear Layer | $N_{\text{dim}} \times 5 + 50$ | $N_{\text{dim}} \times 5 + 50$ |
| Tanh | N/A | N/A |
| Linear Layer | $N_{\text{dim}} \times 5 + 50$ | $N_{\text{dim}} \times 5 + 50$ |
| Tanh | N/A | N/A |
| Linear Layer | $N_{\text{dim}} \times 5 + 50$ | $N_{\text{dim}} \times 5 + 50$ |
| Tanh | N/A | N/A |
| Linear Layer | $N_{\text{dim}} \times 5 + 50$ | $N_{\text{branch}} = 15$ |

Table 4: Network structure of the "trunk" network of the DeepONet in both the linear and semi-linear parabolic caseGoswami et al. [2022].

**Training hyperparameters**   We train both NGO and DeepONet on random PDEs with a defined basis. Specifically, we consider second-order polynomials $\mu_i(x) \in \left\{ \sum_{i=0}^{2} c_i x^i \mid c_i \sim \mathcal{U}(0,1) \right\}$, $x \in \mathbb{R}^{N_{\text{dim}}}$. We set the parameters $r = 0$ and $\sigma = 1$ during training. For each epoch, we sample 6000 initial $x$ values, $x \in [0.1, 0.6]$, and initial $t$ values, $t \in [0, 0.1]$, and PDE parameters $\{c_i\}_{i=0}^{2}$. We calculate PDE solutions $p(x, t)$ through direct Girsanov calculation, NGO, and DeepONet. We use 4000 sample paths for Girsanov calculation NGO. We then minimize the $\ell_1$ loss between Girsanov calculation and NGO or DeepONet using an Adam optimizer, with a learning rate $1 \times 10^{-3}$.

## B.2   SEMI-LINEAR PARABOLIC PDE

**Network structure**   We now provide network structures on the NGO and DeepONet used to calculate solutions of semi-linear parabolic PDEs. For both NGO and DeepONet, we use a network structure that depends on the number of input dimensions, as shown in Tables 3, 4, 5, and 6.

| Operation Layer | Input Channels | Output Channels |
|---|---|---|
| Convolutional Layer | $N_{\text{dim}} \times 2 + 1$ | $N_{\text{dim}} \times 5 + 50$ |
| Softplus | N/A | N/A |
| Convolutional Layer | $N_{\text{dim}} \times 5 + 50$ | $N_{\text{dim}} \times 5 + 50$ |
| Softplus | N/A | N/A |
| Convolutional Layer | $N_{\text{dim}} \times 5 + 50$ | $N_{\text{dim}}$ |

Table 5: Network structure of NGO for estimating the exponential martingale in the semi-linear parabolic case.

**Training hyperparameters**   We train both NGO and DeepONet to estimate random PDEs with a defined basis. Specifically, we consider second-order polynomials $h_i(t, x, s, z) \in \left\{ c_1 \sum \sin(x_i) + c_2 \sum z_i^2 + c_3 \cos(t + s) \mid c_i \sim \mathcal{U}(0,1) \right\}$. We set the parameters $r = 0$ and $\sigma = 1$ during training. For each epoch, we uniformly sample 6000 initial $x$ values, $x \in [0.1, 0.6]$,

| Operation Layer | Input Channels | Output Channels |
|---|---|---|
| Convolutional Layer | $N_{\dim} + 1$ | $N_{\dim} \times 5 + 50$ |
| Softplus | N/A | N/A |
| Convolutional Layer | $N_{\dim} \times 5 + 50$ | $N_{\dim} \times 5 + 50$ |
| Softplus | N/A | N/A |
| Convolutional Layer | $N_{\dim} \times 5 + 50$ | $N_{\dim} \times 5 + 50$ |
| Softplus | N/A | N/A |
| Convolutional Layer | $N_{\dim} \times 5 + 50$ | $N_{\dim} \times 5 + 50$ |
| Softplus | N/A | N/A |
| Convolutional Layer | $N_{\dim} \times 5 + 50$ | $N_{\dim} \times 5 + 50$ |
| Softplus | N/A | N/A |
| Convolutional Layer | $N_{\dim} \times 5 + 50$ | $N_{\dim} \times 5 + 50$ |
| Softplus | N/A | N/A |
| Convolutional Layer | $N_{\dim} \times 5 + 50$ | $N_{\dim}$ |

Table 6: Network structure of NGO for estimating $Z_t$ in the semi-linear parabolic case.

and initial $t$ values, $t \in [0, 0.1]$, and PDE parameters $\{c_i\}_{i=0}^2$. We calculate the PDE solution $p(x, t)$ through direct Girsanov calculation, NGO, and DeepONet. We use 4000 sample paths for Girsanov calculation NGO. We then minimize the $\ell_1$ loss between Girsanov calculation and NGO or DeepONet using the Adam optimizer, with a learning rate of $1 \times 10^{-3}$.

## B.3 GENERATIVE MODELING

**Network structure**   We describe the network structure for the normalizing flow used to model $p_{\mathrm{meta}}$ and the forward SDE. For $p_{\mathrm{meta}}$, we use a real-NVP model [Dinh et al., 2014, 2016] with 32 affine coupling layers, each having the structure as shown in Table 7 for 2-$d$ $p_{\mathrm{meta}}$ and in Table 8 for 100-$d$ $p_{\mathrm{meta}}$.

**Training hyperparameters**   We train the normalizing flows using the "normflows" platform Stimper et al. [2023]. The training dataset for the 2-d and the 100-d case contain 600 samples each. We sample 60 points from each of the 10 Gaussians with different means and standard variances to form the meta-dataset. Training is performed with the Adam optimizer using a learning rate of $5 \times 10^{-4}$ and a weight decay of $1 \times 10^{-5}$. We define the diffusion function of the forward SDE as a $d$-dimensional diagonal matrix, where $d$ is the dimension of the forward SDE. We set the terminal time of the forward SDE as $T = 0.1$, the number of Euler steps when training is 40, and the number of Euler steps when testing is 50. We set the number of samples $N_E$ used to estimate $\mathrm{ELBO_{IS}}$ to be 75. We estimate the divergence with Hutchinson's trace estimator as used in Grathwohl et al. [2018]. We minimize $\mathrm{ELBO_{IS}}$ using an AdamW optimizer and a learning rate of $8 \times 10^{-4}$.

| Operation Layer | Input Number | Output Number |
|---|---|---|
| Linear Layer | 1 | 64 |
| Tanh | N/A | N/A |
| Linear Layer | 64 | 64 |
| Tanh | N/A | N/A |
| Linear Layer | 64 | 64 |
| Tanh | N/A | N/A |
| Linear Layer | 64 | 64 |
| Tanh | N/A | N/A |
| Linear Layer | 64 | 2 |
| Tanh | N/A | N/A |

Table 7: Network structure of the affine coupling layer in the normalizing for 2-d $p_{\mathrm{meta}}$.

| Operation Layer | Input Number | Output Number |
| --- | --- | --- |
| Linear Layer | 50 | 200 |
| Tanh | N/A | N/A |
| Linear Layer | 200 | 200 |
| Tanh | N/A | N/A |
| Linear Layer | 200 | 200 |
| Tanh | N/A | N/A |
| Linear Layer | 200 | 200 |
| Tanh | N/A | N/A |
| Linear Layer | 200 | 100 |
| Tanh | N/A | N/A |

Table 8: Network structure of the affine coupling layer in the normalizing for 100-d $p_{\mathrm{meta}}$.

| Operation Layer | Input Number | Output Number |
| --- | --- | --- |
| Linear Layer | 2 | 200 |
| Tanh | N/A | N/A |
| Linear Layer | 200 | 200 |
| Tanh | N/A | N/A |
| Linear Layer | 200 | 200 |
| Tanh | N/A | N/A |
| Linear Layer | 200 | 200 |
| Tanh | N/A | N/A |
| Linear Layer | 200 | 2 |
| Tanh | N/A | N/A |

Table 9: Network structure of the drift function of the forward SDE in the 2-d case.

| Operation Layer | Input Number | Output Number |
| --- | --- | --- |
| Linear Layer | 100 | 200 |
| Tanh | N/A | N/A |
| Linear Layer | 200 | 200 |
| Tanh | N/A | N/A |
| Linear Layer | 200 | 200 |
| Tanh | N/A | N/A |
| Linear Layer | 200 | 200 |
| Tanh | N/A | N/A |
| Linear Layer | 200 | 100 |
| Tanh | N/A | N/A |

Table 10: Network structure of the drift function of the forward SDE in the 100-d case.

## C  ABLATION STUDY

We perform a series of ablation studies of the proposed NGO algorithm for the linear and semi-linear parabolic PDEs on the number of dimensions of the PDEs and on the number of sample paths used to approximate the solutions.

**Number of dimensions**  We investigate the influence of the number of dimensions on the performance of direct Girsanov calculation (GIR), NGO, DeepONet (DON), and compare to either the analytical solutions or solutions simulated with Euler Maruyama (E-M) using a large number of sample paths.

Figure 5 shows the results on linear parabolic PDEs. As the number of dimensions grows, the normalized errors of NGO increase for random PDEs with a defined basis and decrease in the Fokker-Planck and Black-Scholes equation. The figures suggest that the number of dimensions does not significantly influence the inference time of NGO.

| Operation Layer | Input Number | Output Number |
|---|---|---|
| Linear Layer | 1 | 16 |
| Tanh | N/A | N/A |
| Linear Layer | 16 | 16 |
| Tanh | N/A | N/A |
| Linear Layer | 16 | 16 |
| Tanh | N/A | N/A |
| Linear Layer | 16 | 16 |
| Tanh | N/A | N/A |
| Linear Layer | 16 | 2 |
| Tanh | N/A | N/A |

Table 11: Network structure of the diffusion function of the forward SDE in the 2-d case.

| Operation Layer | Input Number | Output Number |
|---|---|---|
| Linear Layer | 1 | 16 |
| Tanh | N/A | N/A |
| Linear Layer | 16 | 16 |
| Tanh | N/A | N/A |
| Linear Layer | 16 | 16 |
| Tanh | N/A | N/A |
| Linear Layer | 16 | 16 |
| Tanh | N/A | N/A |
| Linear Layer | 16 | 100 |
| Tanh | N/A | N/A |

Table 12: Network structure of the diffusion function of the forward SDE in the 100-d case.

Figure 6 shows the results on semi-linear parabolic PDEs. The number of dimensions does not significantly influence the normalized error of NGO on random PDEs with a defined basis and on the Black-Scholes-Barrenblatt equation. The normalized error of NGO decreases in the Hamilton-Jacobi-Bellman equation as the number of dimensions grows. The inference time of NGO correlates positively with the number of dimensions but is still the lowest among all methods tested.

**Number of sample paths**    We additionally how the number of sample paths used to calculate the solutions correlates with the performance of direct Girsanov calculation, NGO, DeepONet (which is uninfluenced by the number of sample paths), and compare to either the analytical solutions or solutions simulated with Euler Maruyama (E-M) and a large number of sample paths.

Figure 7 shows the results on linear parabolic PDEs. As the number of sample paths grows, the normalized errors of NGO on all three PDEs decrease and then stabilize. This behavior is particularly obvious on random PDEs with a defined basis. Due to parallel rather than sequential computations, the number of sample paths does not significantly influence the inference time of NGO. The number of sample paths does not impact the inference time before reaching the GPU's memory limit.

Figure 8 shows the results on semi-linear parabolic PDEs. As the number of sample paths increases, the normalized error of NGO slightly decreases. The number of sample paths does not significantly influence the normalized errors of NGO on the Hamilton-Jacobi-Bellman equation and the Black-Scholes-Barrenblatt equation. The inference time of NGO first increases with the number of sample paths and then stabilizes but is still the lowest among all methods tested.

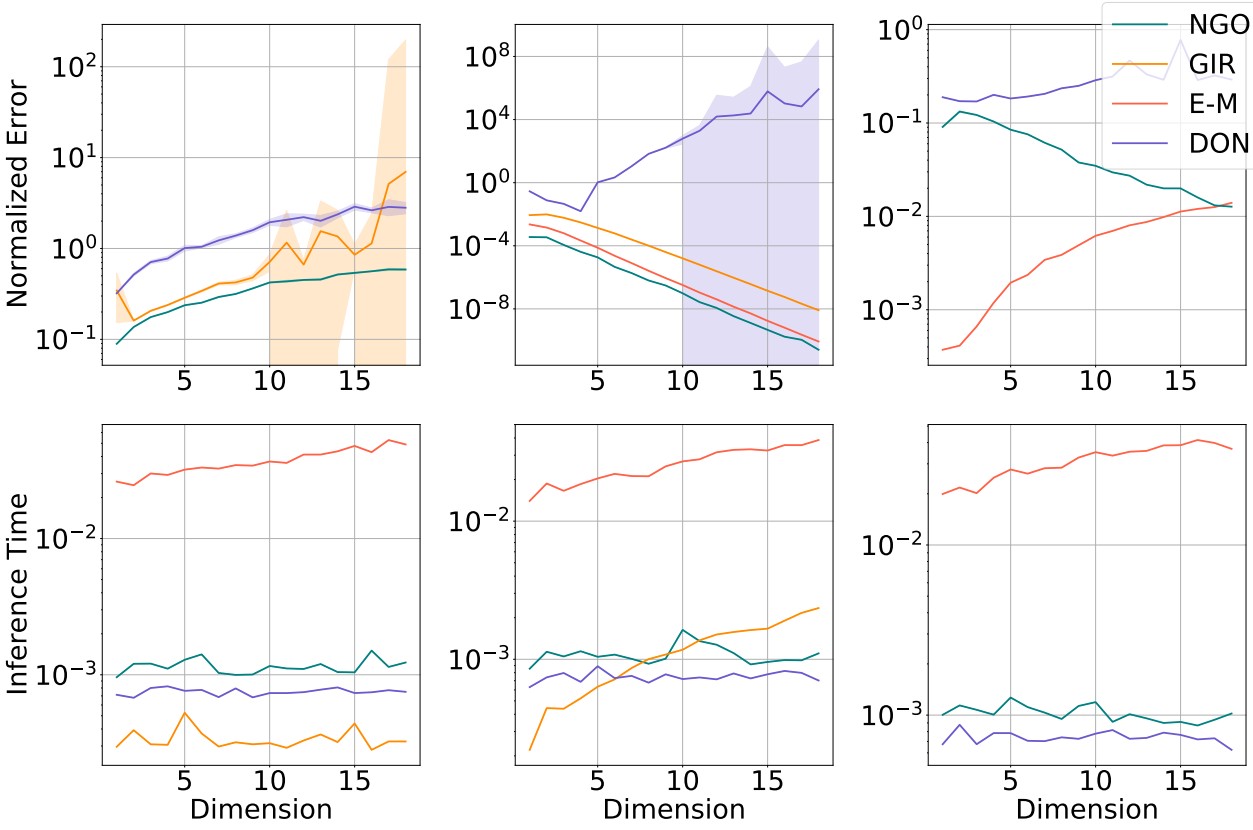

Figure 5: Ablation study on the number of dimensions of linear parabolic PDEs evaluated at the terminal time $T = 0.5$. We show the normalized error (top) and inference time (bottom). The **first column** shows results on random PDEs with a defined basis; the **second column** shows results on the Fokker-Planck equation of the OU process; the **third column** shows results on the Black-Scholes equation.

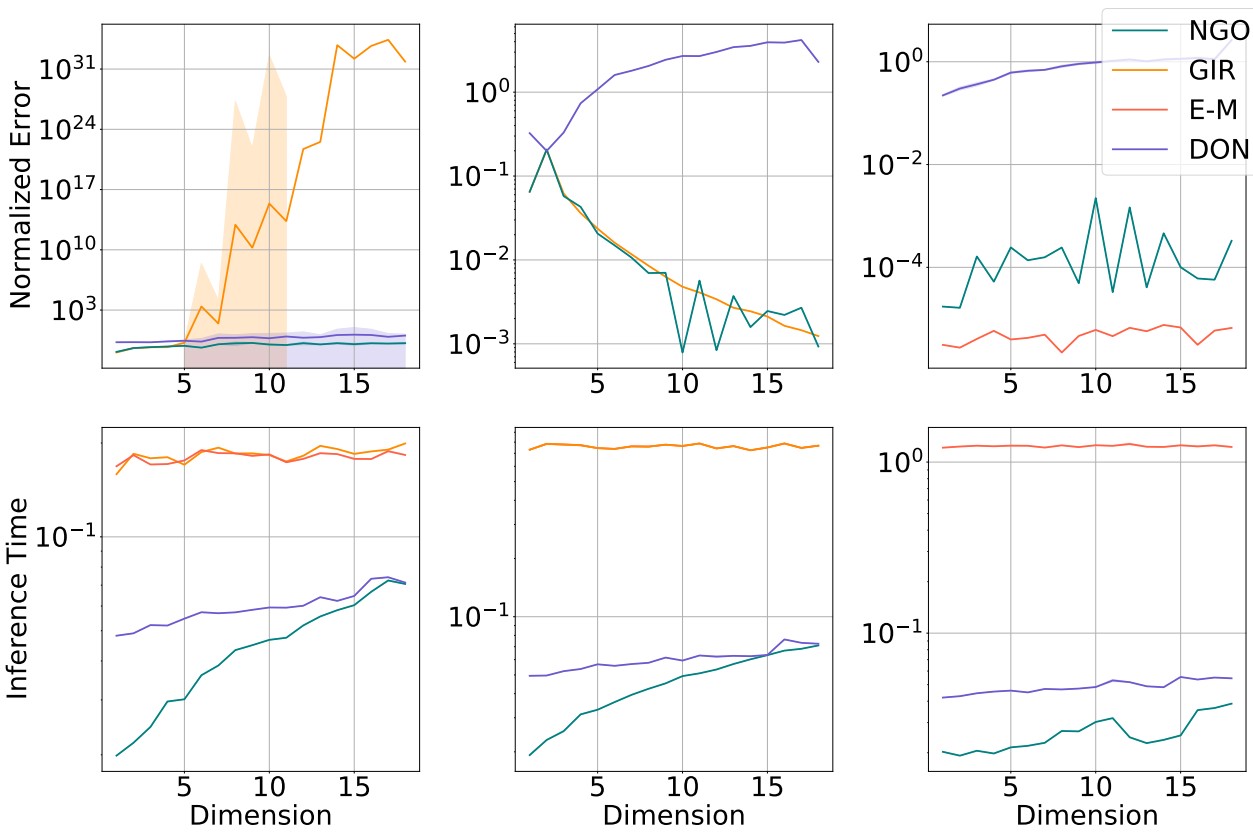

Figure 6: Ablation study on the number of dimensions of semi-linear parabolic PDEs evaluated at terminal time $T = 0.5$. We show the normalized error and inference time. The **first column** shows results on random PDEs with a defined basis; the **second column** shows results on the Hamilton-Jacobi-Bellman equation; the **third column** shows results on the Black-Scholes-Barrenblatt equation. The variance of the Girsanov calculation on the random PDEs with a defined basis increases significantly for $d > 11$.

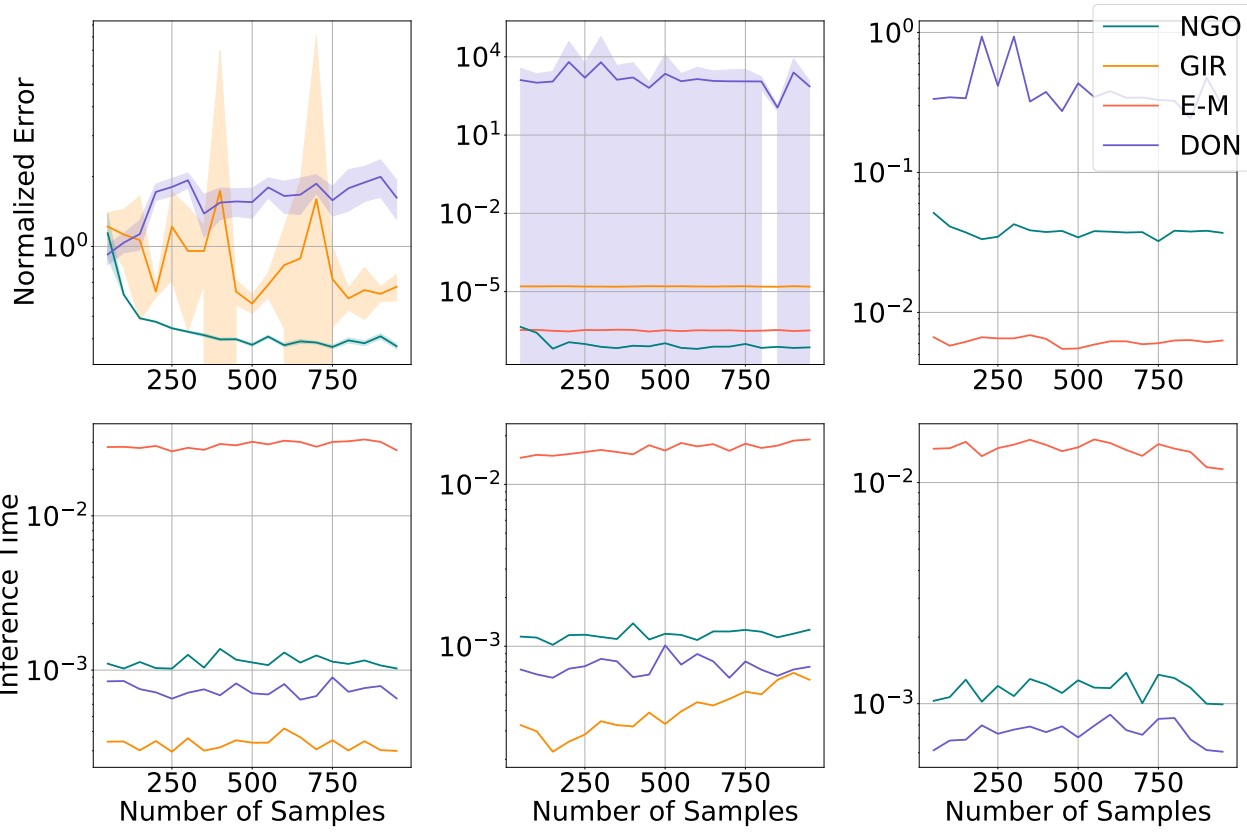

Figure 7: Ablation study on the number of sample paths used in simulating linear parabolic PDEs evaluated at terminal time $T = 0.5$. We show the normalized error (top) and inference time (bottom). The **first column** shows results on random PDEs with a defined basis; the **second column** shows results on the Fokker-Planck equation of the OU process; the **third column** shows results on the Black-Scholes equation.

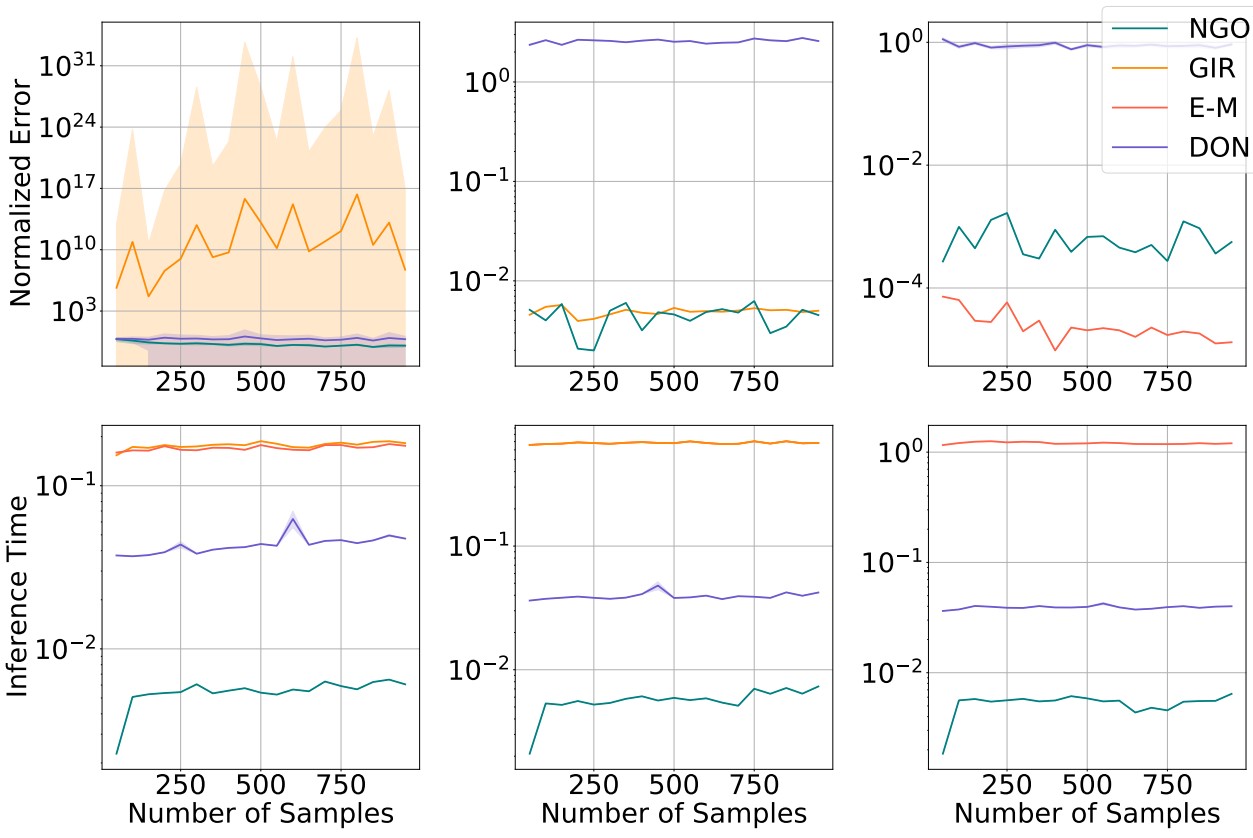

Figure 8: Ablation study on the number of sample paths used in simulating semi-linear parabolic PDEs evaluated at terminal time $T = 0.5$. We show the normalized error (top) and inference time (bottom). The **first column** shows results on random PDEs with a defined basis; the **second column** shows results on the Hamilton-Jacobi-Bellman equation; the **third column** shows results on the Black-Scholes-Barrenblatt equation.

# D DEFINITIONS OF CANONICAL PDES

## D.1 FOKKER PLANCK EQUATION

We study the PDF of a $d$-dimensional time-invariant linear SDE of the form:

$$\begin{cases} \mathrm{d}X_t = X_t \mathrm{d}t + \mathrm{d}W_t, \\ X_0 \sim \mathcal{N}(0, I). \end{cases}$$

$X_t$ follows a Gaussian distribution for all $t$: $X_t \sim \mathcal{N}(m(t), c(t))$, where $m(t)$ and $c(t)$ satisfies the following ODE system:

$$\begin{cases} m(t) &= \exp(t - t_0)m(t_0), \\ c(t) &= \exp(t - t_0)c(t_0)\exp(t - t_0)^T + \int_{t_0}^t \exp(t - \tau)Q\exp(t - \tau)^T d\tau, \end{cases}$$

where $Q$ is the Brownian motion's diffusion coefficient and is assumed to be a $d$-dimensional identity matrix in our case [Särkkä and Solin, 2019].

We assume the mean at initial time $m(t_0) = 0$, and find

$$\begin{cases} m(t) &= 0, \\ c(t) &= (\frac{3}{2}\exp(2t) - \frac{1}{2})I. \end{cases}$$

So we have an analytical form of $X_t$'s distribution: $X_t \sim \mathcal{N}(0, (\frac{3}{2}\exp(2t) - \frac{1}{2})I)$.

Distribution of the SDE $X_t$ corresponds to the initial value problem (IVP) satisfying the Fokker-Planck equation

$$\begin{cases} \frac{\partial p}{\partial t} &= -x \cdot \nabla p + \frac{1}{2}\mathrm{Tr}(\mathrm{Hess}_x p) - dp \\ p_0(x, 0) &= \frac{1}{(2\pi)^d}\exp\left(-\frac{xx^T}{2}\right), \end{cases}$$

where $d$ is the dimension of the SDE, $\cdot$ is inner product, $\times$ is scalar multiplication.

Using the Feynman Kac formula introduced in equation 2, the solution to this IVP has a stochastic representation

$$\begin{cases} p(x, t) &= \exp(-d\,t)\mathbb{E}[p_0(X_t)], \\ \mathrm{d}X_t &= X_t \mathrm{d}t + \mathrm{d}W_t, \\ X_0 &= x. \end{cases}$$

We apply NGO and DeepONet to solve this PDE according to the expectation and compare it with the ground truth PDF.

## D.2 MULTI-DIMENSIONAL BLACK-SCHOLES EQUATION

We consider a multi-variate extension of the Black-Scholes model where multiple, correlated assets govern the price of a derivative. The price evolution of a European call under the Black–Scholes model is modeled by the expectation of the corresponding payoff function with respect to geometric Brownian motion:

$$\begin{cases} \frac{\partial p}{\partial \tau} + \frac{\hat{\sigma}^2}{2}\sum_{i=1}^d s_i^2 \frac{\partial^2 p}{\partial s_i^2} + r(s \cdot \nabla p - p) = 0, \\ p(s, T) = \Phi(s). \end{cases} \tag{13}$$

We apply a change of variable $t = T - \tau$ and transform the terminal condition problem to an equivalent initial value problem (IVP):

$$\begin{cases} \frac{\partial p}{\partial t} = \frac{\hat{\sigma}^2}{2}\sum_{i=1}^d s_i^2 \frac{\partial^2 p}{\partial s_i^2} + r(s \cdot \nabla p - p) = 0, \\ p(s, 0) = \Phi(s). \end{cases} \tag{14}$$

Note that this transformation is not necessary, and the Black-Scholes equation is usually solved according to a terminal condition, but to maintain consistency with the other experiments we consider an IVP. This IVP can be solved using the Feynman-Kac method described in equation 2, with drift $\mu(s) = rs$ and volatility $\sigma(s) = \hat{\sigma}s$ for some prescribed volatility coefficient $\hat{\sigma}$ and payoff $\Phi$.

To get a more accurate estimate, we simplify the terminal value problem 13 with a change of variable by transforming it to a function of Brownian motion, as described in the main text. We will consider the variable $x_i = f(s_i)$ where $f(s) = \ln s + (r - \frac{1}{2}\hat{\sigma}^2)\tau$. Since $\mathrm{d}s_t = rs_t\mathrm{d}t + \hat{\sigma}s_t\mathrm{d}W_t$, applying Itô's lemma to $f(s)$ gives us the new SDE

$$\mathrm{d}f(s_t) = \left( rs_t \underbrace{\left(\frac{1}{s_t}\right)}_{\frac{\partial f}{\partial s}} + \frac{1}{2}\hat{\sigma}^2 s_t^2 \underbrace{\left(\frac{-1}{s_t^2}\right)}_{\frac{\partial f^2}{\partial s^2}} - \underbrace{\left(r - \frac{1}{2}\hat{\sigma}^2\right)}_{\frac{\partial f}{\partial t}} \right)\mathrm{d}t + \hat{\sigma}s_t\left(\frac{1}{s_t}\right)\mathrm{d}W_t$$

$$\mathrm{d}X_t = \hat{\sigma}\mathrm{d}W_t.$$

Now, we can consider taking expectations of $X_t$, where exact sampling is possible since it is Brownian motion. After the transformation, the PDE now corresponds to the heat equation plus the discount factor given by $-rp$:

$$\begin{cases} \frac{\partial \Psi}{\partial t} = \frac{\hat{\sigma}^2}{2}\sum_{i=1}^{N}\frac{\partial^2 \Psi}{\partial x_i^2} - rp, \\ \Psi(x,0) = p(x,0) = \Phi(\exp(x_i + (r - \frac{\hat{\sigma}^2}{2})\tau)). \end{cases} \tag{15}$$

This gives us a relationship between the PDE on the transformed variable and the original one through

$$p(s,t) = \exp(-rt)\Psi\left(\ln s - \left(r - \frac{\hat{\sigma}^2}{2}\right)(T-t), t\right).$$

We take $\Phi(s) = \max\{\max_{k=1\ldots d} s_k - K, 0\}$ for the payoff function, which corresponds to the payoff for the best-asset rainbow option.

We approximate $p(s,t)$ with NGO and DeepONet, following the IVP problem 14, and compare to solutions calculated with the simplified IVP system 15. Note NGO is trained with a constant diffusion term, whereas the SDE system required here has a state-dependent diffusion. The low error of NGO on this problem showcases its generality.

## D.3  HAMILTON-JACOBI-BELLMAN EQUATION

The field of optimal control often requires directly or indirectly solving a terminal value problem involving a $d$-dimensional HJB equation. We study such a problem:

$$\begin{cases} \frac{\partial p}{\partial t} = -\mathrm{Tr}(\mathrm{Hess}_x p) + \|\nabla p\|^2, \\ p(T,x) = g(x), \end{cases}$$

where $g(x) = \ln\left(\frac{1}{2}(1 + \|x\|^2)\right)$.

This PDE has a stochastic solution given by $p(t,x) = -\ln(\mathbb{E}[\exp(-g(x + \sqrt{2}W_{T-t}))])$ [Raissi et al., 2019a].

This PDE's semi-linear results in a representation as a Forward-Backward SDE (FBSDE) of the form of

$$\begin{cases} \mathrm{d}X_t = \sqrt{2}\mathrm{d}W_t, \\ X_0 = 0, \\ \mathrm{d}S_t = \frac{\|Z_t\|^2}{2}\mathrm{d}t + Z_t^T\mathrm{d}W_t, \\ S_T = g(X_T), \end{cases}$$

where $Z_t = \sqrt{2}p_x^t$. We apply NGO on this FBSDE system, with forward drift function $\mu(t,x) = 0$, forward diffusion function $\sigma(t,x) = \sqrt{2}$, backward drift function $h(t,x,p,\sigma^T\nabla_x p) = -\frac{\|\sigma^T\nabla_x p\|^2}{2\sigma^2}$, and terminal condition $g(x) = \ln\left(\frac{1}{2}(1 + \|x\|^2)\right)$. We compare the results generated with NGO, DeepONet, and Girsanov with the analytical solution.

## D.4 BLACK-SCHOLES-BARRENBLATT EQUATION

The Black-Scholes-Barenblatt (BSB) equation is a semi-linear extension to the Black-Scholes equation mentioned in section D.2 and models uncertainty in volatility and interest rates under the Black-Scholes model Li et al. [2019]. We study a terminal value problem involving the BSB equation:

$$\begin{cases} p_t & = -\frac{1}{2}\text{Tr}[\sigma^2 \text{diag}(X_t^2)D^2 p] + r(p - (Dp)^T x), \\ p(T,x) & = \|x\|^2. \end{cases}$$

From Raissi et al. [2019a], this problem has an exact solution given as

$$p(t,x) = \exp((r + \sigma^2)(T-t))g(x). \tag{16}$$

Due to the BSB equation's semi-linear nature, one can represent its solution with an FBSDE system.

$$\begin{cases} dX_t & = \sigma\text{diag}(X_t)dW_t, t \in [0,T], \\ X_0 & = x_0, \\ dS_t & = r(S_t - Z_t^T X_t)dt + \sigma Z_t^T \text{diag}(X_t)dW_t, t \in [0,T), \\ S_T & = g(X_T). \end{cases}$$

Following the FBSDE system, we then construct the NGO, where the forward drift function $\mu(t,x) = 0$, the forward diffusion function $\sigma(t,x) = \sigma\text{diag}(X_t)$, the backward drift function $h(t,x,p,\sigma^T\nabla_x p) = r(p - \frac{\sigma^T \nabla_x p^T}{\sigma x})$, and the terminal condition $g(x) = \|x\|^2$. We compare the explicit solution of the BSB equation with the solutions generated by NGO, DeepONet, and Girsanov.

# E ALGORITHM FOR NGO OF SEMI-LINEAR PARABOLIC PDES

We present the complete algorithm of NGO on semi-linear parabolic PDEs in algorithm 2.

---

**Algorithm 2** Approximating semi-linear PDEs with NGO

---

**Input:** $N \in \mathbb{N}$, $h \in \mathbb{R}_+$, $\mu(t,x) : \mathbb{R}_+ \times \mathbb{R}^d \to \mathbb{R}^d$, terminal time $T$, initial position $X$

1: Sample $N$ Brownian motions to time $T$ starting at $X$, $\left\{ X + \sqrt{kh}\varepsilon^{(i)} \right\}_{k=1...T/h}^{i=1...N}$, $\varepsilon \sim \mathcal{N}(0,1)$

2: **for** $i \in \{1,\dots,N\}$ **do**

3:     Compute $\frac{d\mathbb{P}_\mu^{(i)}}{d\mathbb{P}_W} \approx \text{NGO}^{\text{expmart}}\left[ \left\{ \mu\left(W_k^{(i)}\right) \right\}_{k=1}^{T/h}, \left\{ \sqrt{kh}\varepsilon^{(i)} \right\}_{k=1}^{T/h}, h \right]$

4:     Compute $S_T^{(i)} \approx g(W_T^{(i)})\text{NGO}_T^{\text{expmart}}$ and $Z_T^{(i)} = \text{NGO}^{\text{grad}}\left[ S_T^{(i)}, W_T^{(i)} \right]$

5: **end for**

6: **for** $k \in \{T/h,\dots,1\}$, $i \in \{1,\dots,N\}$ **do**

7:     Compute $S_{T-kh}^{(i)} = S_T^{(i)} + h(T-kh, W_{T-kh}^{(i)}, S_{T-(k-1)h}^{(i)}, Z_{T-(k-1)h}^{(i)}) \times h \times \text{NGO}_{T-(k-1)h}^{\text{expmart}}$

8:     Compute $Z_{T-kh}^{(i)} = \text{NGO}^{\text{grad}}\left[ S_{T-kh}^{(i)}, W_{T-kh}^{(i)} \right]$

9: **end for**

**Output:** Approximation of $u(T-kh, X_{T-kh})$ as $\check{u}(T-kh, X_{T-kh}) = \frac{1}{N}\sum_{i=1}^{N} S_{T-kh}^{(i)}$

---

# F OTHER PARAMETERS FOR META-LEARNING

In the main text, we focused on meta-learning the prior $p_{\text{meta}}$ for the generative modeling task. Here we describe how to meta-learn the other parameters associated with the PDE and provide examples of use cases. In terms of the solution to a PDE, this corresponds to learning an optimal initial condition that satisfies all tasks.

### F.1 META-LEARNING THE BASE $\mu_0$

In the experiments provided in the main text, we always considered sampling from $\mathbb{P}_{X_t}$ being standard Brownian motion (that is, $\mu_0 = 0$). However, this need not be the case. Consider $k$ task-specific $\{\mu_i\}_{i=1}^k$ drifts. We can learn an optimal $\mu_0$ that minimizes the distance between all the task-specific $\mu_i$'s. This has the effect of $\mu_i - \mu_0 \to 0$ for all $i$, which would lead to a However, this comes at the expense of requiring an Euler-Maruyama solve for each iteration of training since $\mu_0$ would need to change.

**Example: baseline policy**   Suppose our interest lies in solving the following maximization problem:

$$\max_{\mu_0} \mathbb{E}_i \left\{ \mathbb{E}_{\mu_i - \mu_0}[J(X_T)] \right\},$$

where we compute the inner expectation over an objective function and the outer expectation over various tasks $i$ with distinct drift functions $\mu_i$. This maximization problem could, for example, relate to the maximization of a portfolio under $k$ different market conditions. The meta-learned parameter $\mu_0$ then describes the optimal policy in all $k$ market conditions. We rewrite this maximization problem with using Girsanov's theorem as

$$\max_{\mu_0} \mathbb{E}_i \left\{ \mathbb{E}_{-\mu_0} \left[ J(X_T) \exp \left( \int_0^T \mu_i dW_t - \frac{1}{2} \int_0^T \mu_i^2 dt \right) \right] \right\},$$

leading to a similar meta-learning problem as described in the main text.

### F.2 META-LEARNING THE BASE $\sigma_0$

In addition to optimizing for the base drift, $\mu_0$, we can also consider optimizing for a baseline $\sigma_0$. Diffusion models utilize similar concepts by learning $\sigma$ values for known SDEs. However, due to challenges with sampling state-dependent diffusion, it is more convenient to consider a function linear in $X_t$. In the following example, we will delve into this issue further.

**Example: baseline volatility**   Suppose our interest lies in sampling from a generative model with distributions that satisfy a Fokker-Planck equation. In order to sample from all target distributions optimized for the considered set of distributions, we can estimate a baseline volatility. Score-based generative modeling applies similar concepts, typically using an affine SDE as a base model that is adapted for different target distributions. To solve the same maximization problem we previously discussed, we can now introduce a parameter $\sigma(t)$ shared among all sample paths for different tasks. We express the problem as:

$$\max_{\sigma_\star(t)} \mathbb{E}_i \left\{ \mathbb{E}_{\mu_0} \log p_0(Y_T) + \int \sigma^{-1}(t) \mu_i dW_t - \frac{1}{2} \int \mu_i^T \sigma^{-1}(t) \mu_i dt \mid Y_0 \sim p_i(Y) \right\}.$$

**Extension to state-dependent volatility**   The primary focus of this study is situations where $\mu$ represents different parameters of a partial differential equation's (PDE) solution or various tasks within a meta-learning framework. We also examine the instances where the volatility, denoted by $\sigma$, changes. We limit our consideration to those volatilities that fulfill the conditions set by the multivariate *Lamperti transform* [Aït-Sahalia, 2008, Proposition 1].

The Lamperti transform imposes specific constraints on the partial derivatives, which come from Itô's Lemma in 2. This property enables converting the corresponding diffusion into a form with unit volatility.

Assuming the existence of a function $f$ such that $\sigma = \nabla_x f(\cdot)$, the PDE can be solved for various parameters using the following approximation:

$$\mathbb{E}_{\mathbb{P}_{X,\sigma}}[p_0(X_T) \mid \mathcal{F}_T] \approx \mathbb{E}_{\mathbb{P}_Y} \left[ p_0(Y_t) \text{NGO} \left( \{\mu(f(Y_{s_n})), \Delta f(W_{s_n}), h\}_{n_T=1}^{N_T}; \theta \right) \mid \mathcal{F}_T \right].$$

Finally, the operator approximates the integral, with the additional component contributed by the trace of the Hessian, as outlined in equation (5).

## G   MAXIMIZATION PROBLEMS

We presented the main motivation of the maximization problem in terms of the Fokker-Planck equation under different types of target distributions.

**Maximizing value functions**    Consider an example where we wish to obtain the policy that maximizes a certain utility function. This situation arises, for instance, in a portfolio optimization problem where we assume a particular stochastic differential equation (SDE) governs a vector of assets $S_t$, and we aim to find the policy $\pi^\star$ that maximizes the utility $J$ across $K$ different market scenarios. In other words, we seek the meta-parameter $\mu_{\text{meta}} = \pi^\star$ that maximizes the utility across the various scenarios, with each scenario specified by the drift $\mu^{(i)}$.

This maximization problem is expressed as $\max_\pi \sum_{i=1}^K \mathbb{E}_{\mu^{(i)}}[J(S_T)]$. To tackle this problem, we employ the same Monte Carlo approach used in the generative modeling case. Assuming a linear interaction between the policy and the assets, we once again apply Jensen's inequality to maximize the expectation of the logarithm. The evidence lower bound (ELBO) in this case is given by:

$$\max_\pi \sum_{i=1}^K \mathbb{E}_\pi \left[ J(S_T) + \int_0^T \mu^{(i)}(S_t)\, \mathrm{d}W_t - \frac{1}{2} \int_0^T \left( \mu^{(i)}(S_t) \right)^2 \mathrm{d}t \right].$$

This formulation eliminates the need to recompute sample paths, as only the sample path corresponding to $\pi$ requires computation.

**Example: Meta-learning $p_0$ for $K$ tasks**    Continuing with our example from the main text, we assume that all tasks have some relationship to each other, and we aim to leverage these relationships to enhance the performance of the sampling task compared to individual training. We consider the initial distribution, $p_0$, as a meta-learned parameter, which we can represent using a parametric model, $p_{\text{meta}}(\,\cdot\,;\theta) \equiv p_0$.

When integrated with the forward stochastic differential equation (SDE), we need to solve for:

$$\max_{p_{\text{meta}}} \sum_{i=1}^K \max_{\mu_i} \text{ELBO}_{\text{IS}}(\{X^{(i)}\}; \mu_i) =$$

$$\max_{p_{\text{meta}}} \sum_{i=1}^K \max_{\mu_i} \mathbb{E} \left[ \int_0^T \mu_i(Y_s, s)\, \mathrm{d}W_s - \frac{1}{2} \int_0^T \mu_i^2(Y_s, s) - \nabla \cdot \mu_i(Y_s, s)\, \mathrm{d}s + \log p_{\text{meta}}(Y_T) \right].$$

By maximizing this expression over all $p_i$ that we want to approximate, with a corresponding $\mu_i$ for each $p_i$, we can use the associated information of each $p_i$ in the form of $p_{\text{meta}}$. It is important to note that the same derivation holds if we compute the expectation over full sample paths without employing importance sampling:

$$\max_{p_{\text{meta}}} \sum_{i=1}^K \max_{\mu_i} \text{ELBO}_{\text{direct}}(\{X^{(i)}\}; \mu_i) = \max_{p_{\text{meta}}} \sum_{i=1}^K \max_{\mu_i} \mathbb{E} \left[ \int_0^T \nabla \cdot \mu_i(X_s^{(i)}, s)\, \mathrm{d}s + \log p_{\text{meta}}\left( X_T^{(i)} \right) \right].$$

However, in the case of $\text{ELBO}_{\text{direct}}$, the sample paths need to be recomputed for each iteration since they depend on the parameters of the drift, whereas the simpler model under $\mathbb{P}_{Y_t}$ needs to be resampled for each iteration of the importance sampling-based method. The direct case requires sequential computation at each time, whereas the computation in $\text{ELBO}_{\text{IS}}$ can be parallelized.

