# OpenReview forum: "Base Models for Parabolic Partial Differential Equations"
_auai.org/UAI/2024/Conference — UAI 2024 poster_

### Official Review · Reviewer_x9i8 · 2024-03-12

**Q2-1 Originality-Novelty:** 3
**Q2-2 Correctness-Technical Quality:** 3
**Q2-5 Clarity Of Writing:** 3

**Q1 Summary And Contributions:**

In this paper, the authors propose a framework for finding solutions to parabolic PDEs across different scenarios by meta-learning an underlying base distribution. They illustrate the application of the proposed methods through experiments in generative modeling, stochastic control, and finance.

**Q2-3 Extent To Which Claims Are Supported By Evidence:**

3: Good: the main claims are supported by convincing evidence (in the form of adequate experimental evaluation, proofs, (pseudo-)code, references, assumptions).

**Q2-4 Reproducibility:**

2: Fair: key resources (e.g. proofs, code, data) are unavailable but key details (e.g. proof sketches, experimental setup) are sufficiently well-described for an expert to confidently reproduce the main results.

**Q3 Main Strengths:**

The authors describe the base model, a corresponding meta-learning framework and a neural operator to approximate the solution. The paper analyzes the convergence of the operator and evaluate the methods in different experimental settings. Theorems are proofed.

The paper was well organized. Overall the paper is sound and the results are presented in an appropriate way. Figures are illustrative and tables are comprehensive. The method was tested on various datasets.

**Q4 Main Weakness:**

Numerical instabilities mentioned in the Limitation section may be a potential issue. I did not see any strong weakness.

**Q5 Detailed Comments To The Authors:**

Same as above.

**Q9 Complying With Reviewing Instructions:**

Yes

---

> ### Author Rebuttal · Authors · 2024-04-05
>
> We thank the reviewer for the encouraging comments and time for providing feedback. Our method is rooted in solving parabolic PDEs by combining the Feynman-Kac method and Girsanov's theorem. Direct application of Girsanov's theorem will cause numerical instabilities due to the exponential term, and the proposed $\mathrm{NGO}$ framework aims to alleviate the instabilities. We will include additional discussion on how the $\mathrm{NGO}$ circumvents these issues.

---

### Official Review · Reviewer_Y66b · 2024-03-22

**Q2-1 Originality-Novelty:** 2
**Q2-2 Correctness-Technical Quality:** 3
**Q2-5 Clarity Of Writing:** 2

**Q10 Ethical Concerns:**

None that I could see.

**Q1 Summary And Contributions:**

A meta-learning method to accelerate the solution of parabolic Partial Differential Equations.

The method builds over an equivalence between the solution of parabolic PDEs and the expectation of certain stochastic processes; the link is exploited to simplify the computation by: 1) sampling a simpler form of stochastic process (Brownian motion), so as to obtain a "meta-distribution"; 2) relying on importance sampling to adapt the meta-distribution to different parameters, so that boundary condition problems unseen at "training" tie can be solved.

The computation of the importance sampling probabilities naturally involves the exponential of numerically integrated integrals, which can be prone to errors. The authors improve the method stability by acknowledging that at training time the probability ratio should be maximized (as part of maximum likelihood estimation) and then by focusing on a evidence base lower bound obtained via Jensen's inequality.

Finally, the authors attempt to learn a ML model (referred to as neural operator) to directly transform the reference stochastic process into parabolic PDE solution, thus bypassing some of the previously described steps.

**Q2-3 Extent To Which Claims Are Supported By Evidence:**

3: Good: the main claims are supported by convincing evidence (in the form of adequate experimental evaluation, proofs, (pseudo-)code, references, assumptions).

**Q2-4 Reproducibility:**

2: Fair: key resources (e.g. proofs, code, data) are unavailable but key details (e.g. proof sketches, experimental setup) are sufficiently well-described for an expert to confidently reproduce the main results.

**Q3 Main Strengths:**

The paper presents two integration approaches for partial PDEs that relies on off-line steps (sampling from the reference process, neural operator learning) to reduce online computational costs. The methods rely on sound intuition and, to the best of my limited understanding, appear to be correct.

The computational experiments provide evidence for the method effectiveness on classical stochastic PDEs.

**Q4 Main Weakness:**

The work is very dense and hard to approach for readers lacking an extensive background on PDEs; the work is also not self contained, with several references to external seminal sources, provided with limited context. Nevertheless, the presentation is however structured well enough to allow even a non-expert to follow the approach rationale.

It is not entirely clear how the operator learning approach is tied to the previous presentation. Intuitively, learning an operator could be done also based on examples from conventional solution approaches (e.g. Euler-Maruyama with a very small integration step). This appears to be mentioned in the conclusions. While the training time would be longer using E-M, the resulting operator would be equally valid. Since the NGO (operator learning) approach is the one actually employed in the empirical evaluation, the significance of the contributions from the first part of the work appear unclear.

**Q5 Detailed Comments To The Authors:**

* SPD_d not defined after equation 1 (I guess it's the space of Semi Definite Positive matrices)
* r(X_sI) not defined in equation 3
* Allowing public access to the code in case of acceptance would considerably help with reproducibility

**Q9 Complying With Reviewing Instructions:**

Yes

---

> ### Author Rebuttal · Authors · 2024-04-05
>
> We thank the reviewer for the  suggestions and time in constructing the review.
> ### Including references
> We apologize for the confusion here and will include a few more references to improve the self-containment of the work.
> Specifically, we will include [1], which provides thorough treatments of the Feynman-Kac method and Girsanov's theorem.
>
> [1] Simo Särkkä and Arno Solin. Applied stochastic differential equations, volume 10. Cambridge University Press, 2019.
>
> ### Operator learning
> We describe a neural operator architecture based on expanding the Feynman-Kac method. In particular, while the Feynman-Kac method calls for simulating new SDE paths with Euler-Maruyama when solving new parabolic PDEs, $\mathrm{NGO}$ attempts to reuse existing paths simulated from Euler-Maruyama. We still obtain solutions to PDEs via the Feynman-Kac method, only including an additional likelihood ratio calculated with $\mathrm{NGO}$.
> Readers can view $\mathrm{NGO}$ as a particular type of neural network architecture that expands the expectation used in the Feynman-Kac method, which is reusable for different parameter regimes.
>
> ### Detailed comments
> We appreciate the suggestions. We will add the definitions and publicize the code in case of acceptance.

---

### Official Review · Reviewer_e2cc · 2024-03-23

**Q2-1 Originality-Novelty:** 3
**Q2-2 Correctness-Technical Quality:** 2
**Q2-5 Clarity Of Writing:** 2

**Q1 Summary And Contributions:**

The authors propose a meta-learning method to solve parabolic partial differential equations. Solving such parabolic PDEs are important as they appear in various scenarios. Current methods for solving these parabolic PDEs require Monte Carlo sampling, and this sampling needs to be done from scratch for every initial condition, boundary condition. This can be time consuming. The authors proposed meta-learning an underlying base distribution, and use this base distribution to solve PDE for different initial/boundary condition. Experimental results show that authors proposed NGO learning method provides better performances in terms of accuracy and computation time, than the baselines.

**Q2-3 Extent To Which Claims Are Supported By Evidence:**

3: Good: the main claims are supported by convincing evidence (in the form of adequate experimental evaluation, proofs, (pseudo-)code, references, assumptions).

**Q2-4 Reproducibility:**

2: Fair: key resources (e.g. proofs, code, data) are unavailable but key details (e.g. proof sketches, experimental setup) are sufficiently well-described for an expert to confidently reproduce the main results.

**Q3 Main Strengths:**

The authors provide good motivation for their work, detailing its importance, providing related citations to put their work in context. The experimental results are good. The authors also mention the limitation scenario of their work clearly, which is a plus.

**Q4 Main Weakness:**

The paper has several scopes for improvement that the authors can consider:
1. The formulation of parabolic PDEs in equation 1 is significantly more complex commonly available ones (for example wiki), can the authors please provide a reference for this formulation, or a short explanation justifying this formulation?
2. What are the parameters of the PDE in equation 1?
3. The key takeaways for figure 1 needs to be communicated more clearly. It is not clear what is happening in figure 1a, please describe the 6 symbols in the figure legend, what is the task, and how meta-learning works here. This would greatly increase the clarity. For figure 1b, perhaps showing the ground truth and the meta-learned solution will be more convincing to the readers that the authors' method does in fact provide good PDE solutions.

**Q5 Detailed Comments To The Authors:**

The paper is a bit hard to follow for people with general understanding of PDEs. Perhaps improving on the figure-1, and adding an example figure showing the authors' method illustratively will help the authors to attract a broader audience.

**Q9 Complying With Reviewing Instructions:**

Yes

---

> ### Author Rebuttal · Authors · 2024-04-05
>
> We thank the reviewer for the encouraging words and thoughtful suggestions.
> ### Formulation of parabolic PDEs
> We appreciate the reviewer for the suggestion. We will add a citation to equation (7.105) in [1]. The formulation also appears in the Wikipedia page for the Feynman-Kac method [2].
>
> [1] Simo Särkkä and Arno Solin. Applied stochastic differential equations, volume 10. Cambridge University Press, 2019.
>
> [2] https://en.wikipedia.org/wiki/Feynman%E2%80%93Kac_formula
>
> ### Parameters of PDE in equation 1
> We mainly investigated $\mathrm{NGO}$'s performances across different drift functions $\ mu$ since these are often the parameters of interest in many applied settings.
> For example, in the Black-Scholes equation, changing $\mu$ corresponds to changes in interest rates, which often need to be recalibrated.
> Additionally, we can easily achieve paths of different volatility functions $\sigma$ by applying It\^o's lemma, as pointed out in the appendix. In particular, we parameterized the function $\mu$ in equation 1 as a polynomial $\mu(x) = \sum_{i=0}^2c_i x^i$. Changing the three $c_i$ results in different $\mu$, which results in a new solution to the PDE under different parameters $\mu$.
>
> ### Figure 1
> We appreciate the reviewer's suggestions for improving Figure 1. Figure 1(a) illustrates the task of generative modeling based on the Fokker-Planck equation, discussed in section 3.2. In particular, the grey plots here are the meta-learned base distribution $p_{\mathrm{meta}}$, which is the initial distribution, and the blue and the green are two different subtasks, with the blue being transforming the base distribution to two Gaussian distributions $p_{\mathrm{Gaussians}}$, and the green being transforming to a two moons distribution $p_{\mathrm{Moons}}$. $Y_t^{\mathrm{meta}}$ is the base SDE, transforming a standard Gaussian distribution to $p_{\mathrm{meta}}$, and $X_t^{\mathrm{Gaussians}}$ and $X_t^{\mathrm{Moons}}$ are SDEs transforming $p_{\mathrm{meta}}$ to $p_{\mathrm{Gaussians}}$ and $p_{\mathrm{Moons}}$.

---

### Official Review · Reviewer_TAYA · 2024-03-24

**Q2-1 Originality-Novelty:** 3
**Q2-2 Correctness-Technical Quality:** 3
**Q2-5 Clarity Of Writing:** 3

**Q1 Summary And Contributions:**

The paper proposes a meta-learning framework for parabolic PDEs of certain type. Specifically, given a stochastic ordinary differential equation (SDE) with some parameters $\mu$ and $\sigma$ one can compute the solution of another SDE by reusing the same paths (without resampling them). This is based on the so-called Girsanov theorem. Specifically, a neural operator is learned to map coefficients and Brownian motion to the solution.

**Q2-3 Extent To Which Claims Are Supported By Evidence:**

3: Good: the main claims are supported by convincing evidence (in the form of adequate experimental evaluation, proofs, (pseudo-)code, references, assumptions).

**Q2-4 Reproducibility:**

3: Good: key resources (e.g. proofs, code, data) are available and key details (e.g. proofs, experimental setup) are sufficiently well-described for competent researchers to confidently reproduce the main results.

**Q3 Main Strengths:**

Writing and idea looks interesting to me.
Theoretical analysis is attempted

**Q4 Main Weakness:**

I do not see clear connection between the first part (in relation to Girsanov theorem) and the algorithm.

Numerical experiments are not fully convincing.1

**Q5 Detailed Comments To The Authors:**

1. What is the exact connection between the Girsanov theorem and the proposed algorithm? To me it seems like a mapping between dW and the solution, i.e. standard neural operator setting
2. I don't fully understand Table 1. Why FNO is a baseline?

**Q9 Complying With Reviewing Instructions:**

Yes

---

> ### Author Rebuttal · Authors · 2024-04-05
>
> We thank the reviewer for the time and effort in providing feedback for this paper.
>
> ### Connection between Girsanov's theorem and the algorithm
>
> We apologize for the confusion here. The neural operator uses Girsanov's theorem to change the drift parameter of the PDE. Naively approximating the Girsanov term results in large numerical instabilities; therefore, our proposed $\mathrm{NGO}$ network learns the likelihood ratio $\frac{d\mathbb{P}_X}{d\mathbb{P}_Y}$ between the origin measure $\mathbb{P}_X$ and the target measure $\mathbb{P}_Y$. This formulation allows us to change the input of the $\mathrm{NGO}$ to solve new PDEs with different parameters without requiring regenerating new sample paths, which is computationally expensive.
>
>
> ### FNO baselines
> FNO is a standard benchmark when considering neural operators for approximating PDEs.
> We chose it due to it being well-known in the literature and it is a high performing algorithm with which we can compare the performance of the proposed method.
> Our comparison with FNO aims to show that $\mathrm{NGO}$ achieves similar losses and faster inference times on applicable PDEs.

---

### Meta-Review · Area_Chair_KWGH · 2024-04-16

All Reviewers agreed to accept this paper, who appreciate that this paper (1) proposes interesting idea with theoretical analysis, (2) good motivation, well positioned in the literature, (3) the algorithm is well grounded by experiments, etc. I agree with the consensus that the Reviewers achieved.